# Differential requirements of androgen receptor in luminal progenitors during prostate regeneration and tumor initiation

Chee Wai Chua[1,2,3,4,5†‡], Nusrat J Epsi[6,7], Eva Y Leung[1,2,3,4,5],
Shouhong Xuan[1,2,3,4,5], Ming Lei[1,2,3,4,5], Bo I Li[1,2,3,4,5], Sarah K Bergren[1,2,3,4,5],
Hanina Hibshoosh[5,8], Antonina Mitrofanova[6,7], Michael M Shen[1,2,3,4,5]*

[1]Department of Medicine, Columbia University Medical Center, New York, United States; [2]Department of Genetics and Development, Columbia University Medical Center, New York, United States; [3]Department of Urology, Columbia University Medical Center, New York, United States; [4]Department of Systems Biology, Columbia University Medical Center, New York, United States; [5]Herbert Irving Comprehensive Cancer Center, Columbia University Medical Center, New York, United States; [6]Department of Health Informatics, Rutgers School of Health Professions, Rutgers, The State University of New Jersey, Newark, United States; [7]Rutgers Biomedical and Health Sciences, Rutgers, The State University of New Jersey, Newark, United States; [8]Department of Pathology and Cell Biology, Columbia University Medical Center, New York, United States

*For correspondence:
mshen@columbia.edu

Present address: †State Key Laboratory of Oncogenes and Related Genes, Renji-Med X Clinical Stem Cell Research Center, Ren Ji Hospital, School of Medicine, Shanghai Jiao Tong University, Shanghai, China; ‡Department of Urology, Ren Ji Hospital, School of Medicine, Shanghai Jiao Tong University, Shanghai, China

Competing interests: The authors declare that no competing interests exist.

**Abstract** Master regulatory genes of tissue specification play key roles in stem/progenitor cells and are often important in cancer. In the prostate, androgen receptor (AR) is a master regulator essential for development and tumorigenesis, but its specific functions in prostate stem/progenitor cells have not been elucidated. We have investigated AR function in CARNs (CAstration-Resistant Nkx3.1-expressing cells), a luminal stem/progenitor cell that functions in prostate regeneration. Using genetically–engineered mouse models and novel prostate epithelial cell lines, we find that progenitor properties of CARNs are largely unaffected by AR deletion, apart from decreased proliferation *in vivo*. Furthermore, AR loss suppresses tumor formation after deletion of the *Pten* tumor suppressor in CARNs; however, combined *Pten* deletion and activation of oncogenic *Kras* in AR-deleted CARNs result in tumors with focal neuroendocrine differentiation. Our findings show that AR modulates specific progenitor properties of CARNs, including their ability to serve as a cell of origin for prostate cancer.
DOI: https://doi.org/10.7554/eLife.28768.001

## Introduction

Elucidating the cell type(s) of origin of cancer and the molecular drivers of tumor initiation is of fundamental importance in understanding the basis of distinct tumor subtypes as well as differences in treatment response and patient outcomes (*Blanpain, 2013*; *Rycaj and Tang, 2015*; *Shibata and Shen, 2013*; *Visvader, 2011*). Furthermore, since cancer often originates from stem cells and/or lineage-restricted progenitor cells, the identification of stem/progenitor cells is of considerable significance. In the case of the prostate, however, both the specific identity of stem/progenitor cells as

**eLife digest** Most prostate tumors rely on male hormones – called androgens – to survive. Aggressive prostate cancer is often treated with drugs that block androgens, which usually cause the prostate tumors to shrink. One class of the drugs works by binding to and inactivating the androgen receptor protein on prostate cancer cells. However, aggressive prostate tumors can often become resistant to these anti-androgen therapies.

It is not clear where the resistant cancer cells come from. In 2009, researchers showed that the normal prostate contains some cells that appear to be independent of androgens. A subset of these cells – also known as CARNs – can act as stem or progenitor cells that can repair the prostate after injury. These normal androgen-independent cells can also be the cells from which prostate tumors arise. Here, Chua et al. – including one of the researchers from the 2009 study – investigated how these CARN cells behave when the androgen receptor is deleted.

When the androgen receptor was genetically removed in CARN cells of otherwise healthy mice, the behavior of CARN cells was unaffected. When the androgen receptor was deleted together with a protein that normally suppresses the formation of tumors, it protected the mice from prostate cancer. However, Chua et al. also observed that deleting the androgen receptor could not prevent the tumor from growing when two cancer-causing mutations were present. These tumors were similar to human prostate tumors that are resistant to anti-androgen therapy.

Since CARN cells may also exist in humans, this new way of making prostate cancers in mice may be used to study how these resistances arise in patients. A better understanding of how prostate tumors develop might lead to new treatments in which the androgen receptor is blocked in combination with other new protein targets.

DOI: https://doi.org/10.7554/eLife.28768.002

well as cell types of origin for cancer have remained unclear (*Lee and Shen, 2015*; *Wang and Shen, 2011*; *Xin, 2013*).

In the normal prostate epithelium, there are three primary cell types, corresponding to secretory luminal cells, an underlying layer of basal cells, and rare neuroendocrine cells (*Shen and Abate-Shen, 2010*; *Toivanen and Shen, 2017*). Lineage-tracing studies have shown that both luminal and basal cells are mostly lineage-restricted (unipotent) in the normal adult mouse prostate as well as during androgen-mediated prostate regeneration (*Choi et al., 2012*; *Liu et al., 2011*; *Lu et al., 2013*; *Wang et al., 2013*). In addition, cells within the basal compartment possess stem/progenitor properties in a range of ex vivo assays as well as during inflammation and wound repair (*Goldstein et al., 2008*; *Höfner et al., 2015*; *Kwon et al., 2014*; *Lawson et al., 2007*; *Toivanen et al., 2016*; *Wang et al., 2013*). However, recent studies have shown that luminal cells can also display stem/progenitor properties in specific *in vivo* and *ex vivo* contexts (*Chua et al., 2014*; *Karthaus et al., 2014*; *Kwon et al., 2016*; *Wang et al., 2009*). Furthermore, there is now considerable evidence supporting a luminal origin for prostate cancer, both in mouse models (*Wang et al., 2009*; *Wang et al., 2014*) as well as in human tissues (*Gurel et al., 2008*; *Meeker et al., 2002*).

Androgen receptor (AR) plays a central role in many aspects of normal prostate development as well as prostate cancer progression (*Cunha et al., 2004*; *Toivanen and Shen, 2017*; *Watson et al., 2015*). In the prostate epithelium of adult hormonally intact mice, AR is primarily expressed by luminal cells, but is also found in a subset of basal cells (*Lee et al., 2012*; *Mirosevich et al., 1999*; *Xie et al., 2017*). Several studies have shown that conditional deletion of AR in the adult prostate epithelium results in a short-term increase in proliferation of luminal cells (*Wu et al., 2007*; *Xie et al., 2017*; *Zhang et al., 2016a*), indicating a role for AR in normal prostate homeostasis. Importantly, AR can act as a master regulator of prostate epithelial specification in a fibroblast reprogramming assay (*Talos et al., 2017*).

In the context of prostate cancer, tumor recurrence after androgen-deprivation therapy is due to the emergence of castration-resistant prostate cancer (CRPC), which is associated with increased AR activity that can be targeted by second-generation anti-androgen therapies (*Watson et al., 2015*). However, treatment failure following such anti-androgen therapies is frequently associated with the

appearance of AR-negative tumor cells, which are typically associated with highly aggressive lethal disease (*Beltran et al., 2014*; *Vlachostergios et al., 2017*; *Watson et al., 2015*). In some cases, this AR-negative CRPC contain large regions displaying a neuroendocrine phenotype (CRPC-NE) (*Beltran et al., 2016*, *2014*; *Ku et al., 2017*; *Mu et al., 2017*; *Zou et al., 2017*).

Previous work from our laboratory has identified CARNs as a luminal stem/progenitor cell within the androgen-deprived normal mouse prostate epithelium that is also a cell of origin for prostate cancer (*Wang et al., 2009*). Following androgen administration to induce prostate regeneration, CARNs can generate both luminal and basal progeny *in vivo*, as well as in renal grafting and organoid assays (*Chua et al., 2014*; *Wang et al., 2009*). Although CARNs express AR (*Wang et al., 2009*), it has been unclear whether AR is required for any or all the progenitor properties of CARNs, and whether the intrinsic castration-resistance of untransformed CARNs might resemble the castration-resistance of tumor cells in CRPC. Below, we show that the progenitor properties of CARNs are largely unaffected by loss of AR, whereas their ability to serve as cells of origin for prostate cancer are altered by AR deletion in a context-dependent manner. Notably, cell lines derived from AR-deleted CARNs have molecular profiles that resemble those for CRPC, and AR-deleted CARNs can serve as a cell of origin for focal neuroendocrine differentiation in a novel mouse model of AR-negative prostate cancer.

## Results

To investigate whether the stem/progenitor properties of CARNs are dependent upon AR function, we have used an inducible targeting approach in genetically engineered mice. For this purpose, we used mice carrying a conditional allele of *Ar* (*De Gendt et al., 2004*) together with the inducible *Nkx3.1*$^{CreERT2}$ driver (*Wang et al., 2009*) and the *R26R-YFP* reporter to visualize cells and their progeny in which Cre-mediated recombination has taken place (*Srinivas et al., 2001*); as *Ar* is an X-linked gene, deletion of a single allele in males is sufficient to confer a hemizygous null phenotype. Since CARNs are Nkx3.1-expressing cells found under androgen-deprived conditions, we castrated adult male mice carrying the Cre driver and reporter alleles, followed by tamoxifen induction to induce Cre-mediated activity specifically in CARNs (*Figure 1A*).

Using this strategy, we compared the properties of CARNs in *Nkx3.1*$^{CreERT2/+}$; *R26R-YFP/+* mice, which we denote as 'control' mice, with those in *Nkx3.1*$^{CreERT2/+}$; *Ar*$^{flox/Y}$; *R26R-YFP/+* mice, which we denote as 'AR-deleted' mice. We found that the percentage of lineage-marked YFP-positive cells, corresponding to CARNs, was not significantly different (p=0.51) between the control (0.36 ± 0.17%, n = 5 mice) and AR-deleted mice (0.31 ± 0.06%, n = 5 mice) (*Figure 1B,C*). Notably, we found that 87.1% of the YFP-positive cells in *Nkx3.1*$^{CreERT2/+}$; *Ar*$^{flox/Y}$; *R26R-YFP/+* mice (n = 344/395 cells in four mice) were AR-negative, indicating that AR deletion occurred with high efficiency. Furthermore, these YFP-positive cells expressed the luminal markers cytokeratins 8 and 18 (CK8 and CK18), but not cytokeratin 5 (CK5) and p63, indicating that AR deletion does not alter the luminal phenotype of CARNs (*Figure 1D*). These findings indicate that AR deletion does not affect the frequency or luminal properties of CARNs.

To investigate the progenitor properties of AR-deleted CARNs, we examined their ability to generate progeny during androgen-mediated regeneration. We implanted subcutaneous mini-osmotic pumps containing testosterone into control *Nkx3.1*$^{CreERT2/+}$; *R26R-YFP/+* mice as well as *Nkx3.1*$^{CreERT2/+}$; *Ar*$^{flox/Y}$; *R26R-YFP/+* mice, followed by tissue harvest at 4, 7, 14, and 28 days later; the final 28-day time point corresponds to a fully regenerated prostate (*Figure 2A*). We found that the YFP-marked cells and cell clusters were similar in the control and AR-deleted prostates at 4 and 7 days after testosterone administration (*Figure 2B,C*). However, at 14 and 28 days, the control prostates contained many YFP-expressing cell clusters with more than 4 cells, whereas the prostates with AR-deleted CARNs mostly contained YFP-expressing single cells or doublets (*Figure 2B,C*).

To compare the proliferative ability of control and AR-deleted CARNs and their progeny, we pursued BrdU pulse-chase experiments during prostate regeneration. We performed castration and tamoxifen administration on control and AR-deleted mice, followed by androgen-mediated regeneration for 28 days, with administration of daily doses of BrdU either from days 1 through 4 of regeneration or from days 11 through 14 (*Figure 3A,B*). When BrdU was administered from days 1 through 4 of regeneration, we could readily detect BrdU$^+$YFP$^+$ cells in the control prostates (50.9 ± 11.8%, n = 3 mice) as well as AR-deleted prostates (62.9 ± 14.9%, n = 3 mice) (*Figure 3C,E*). In contrast,

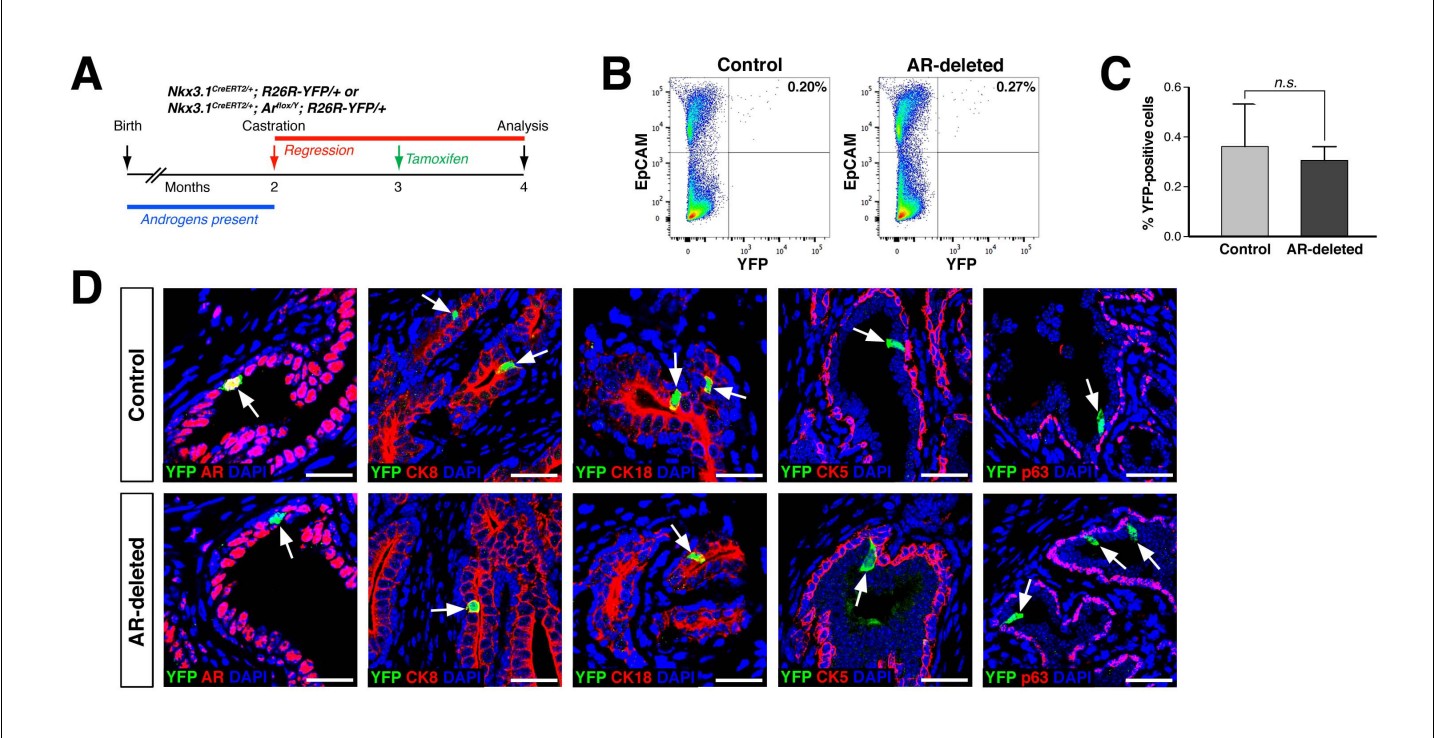

**Figure 1.** CARNs remain luminal after AR deletion. (**A**) Time course for lineage-marking of CARNs and inducible *AR* deletion using castrated and tamoxifen-treated control *Nkx3.1^{CreERT2/+}; R26R-YFP/+* mice and *Nkx3.1^{CreERT2/+}; Ar^{flox/Y}; R26R-YFP/+* mice. (**B**) FACS analyses of lineage-marked YFP⁺ cells in total EpCAM⁺ epithelial cells. (**C**) Percentage of YFP⁺ cells among total epithelial cells in castrated and tamoxifen-induced *Nkx3.1^{CreERT2/+}; R26R-YFP/+* controls and *Nkx3.1^{CreERT2/+}; Ar^{flox/Y}; R26R-YFP/+* mice. Error bars represent one standard deviation; the difference between groups is not significant (p=0.51, independent t-test). (**D**) Expression of AR, luminal markers (CK8 and CK18), and basal markers (CK5 and p63) in lineage-marked CARNs (top) and AR-deleted CARNs (bottom). Note that all lineage-marked cells express luminal but not basal markers (arrows). Scale bars in **D**) correspond to 50 μm.

DOI: https://doi.org/10.7554/eLife.28768.003

The following source data is available for figure 1:

**Source data 1.** Quantitation of CARNs and AR-deleted CARNs *in vivo*.
DOI: https://doi.org/10.7554/eLife.28768.004

when BrdU was administered from days 11 through 14, we could only detect BrdU⁺YFP⁺ cells in the control prostates (11.1 ± 6.2%, n = 3 mice), but not in the AR-deleted prostates (0%, n = 3 mice) (*Figure 3D,F*). This difference suggests that AR-deleted CARNs and/or their progeny have a defect in proliferation during later stages of regeneration, consistent with the analysis of YFP⁺ cluster size (*Figure 2B*).

Notably, although YFP-expressing basal cells could be readily identified in both control and AR-deleted prostates, there was an increase in the percentage of basal cells within the YFP⁺ population in the AR-deleted mice (*Figure 2D*). This difference was evident using either the basal marker CK5 (2.1% CK5⁺AR⁺YFP⁺ versus 19.2% CK5⁺AR⁻YFP⁺) or p63 (3.5% p63⁺AR⁺YFP⁺ versus 14.6% p63⁻⁺AR⁻YFP⁺) (*Figure 2D*). These findings indicate that AR-deleted CARNs favor generation of basal progeny and/or that there is decreased proliferation or survival of luminal progeny during regeneration.

As a further test of the progenitor properties of AR-deleted CARNs, we examined their ability to generate prostate ducts in a tissue recombination/renal grafting assay. Previously, we had shown that single CARNs were capable of generating ducts in this assay (*Wang et al., 2009*). We isolated YFP-positive cells from control and AR-deleted mice that had undergone castration and tamoxifen induction, and recombined 10 YFP-positive cells together with 2.5 × 10⁵ rat embryonic urogenital mesenchyme cells, followed by renal grafting (*Figure 3G*). We found that both control and AR-deleted CARNs could generate prostate ducts (*Figure 3H*), but that the AR-deleted CARN-s were

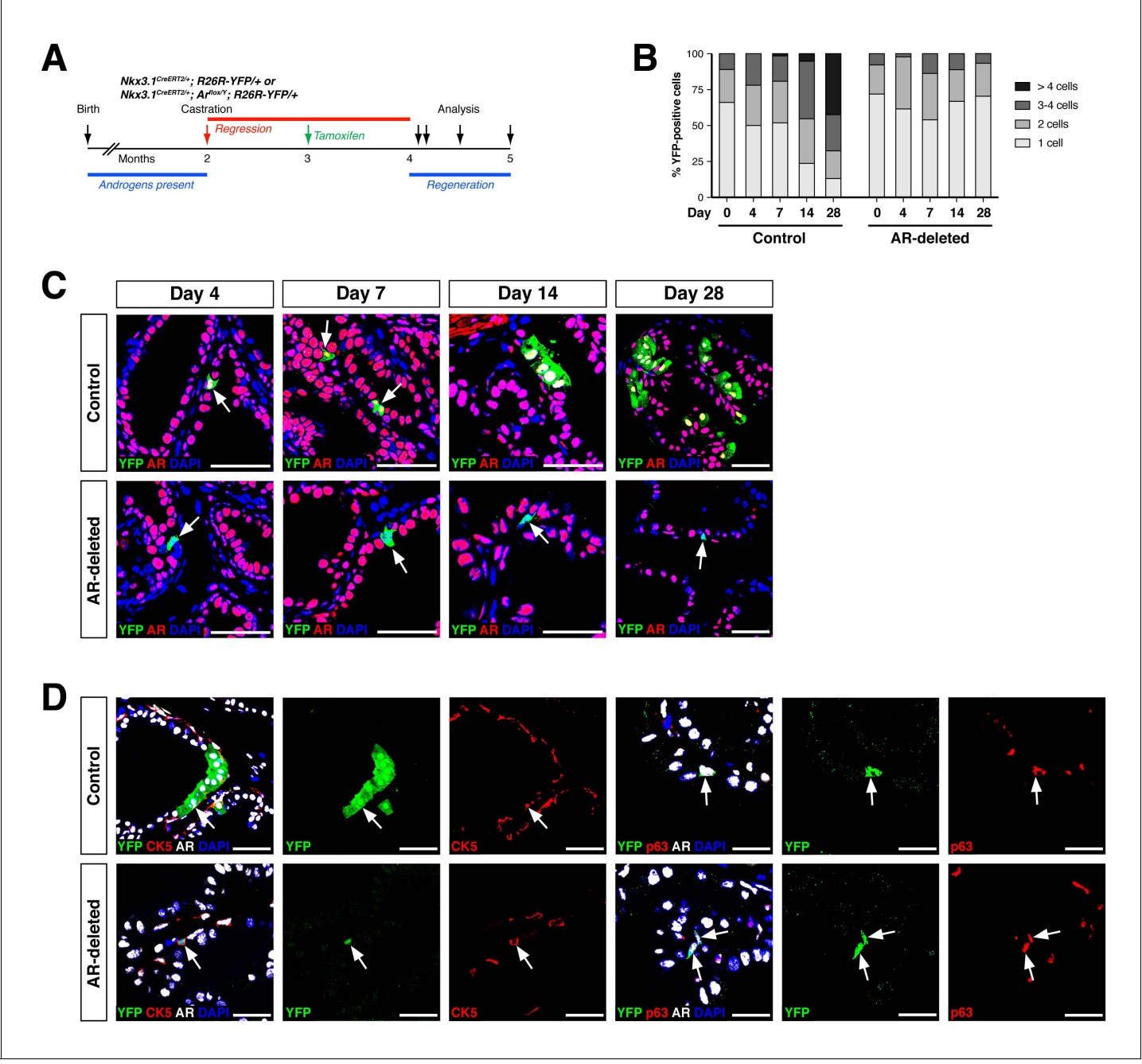

**Figure 2.** AR-deleted CARNs fail to generate lineage-marked cell clusters but remain bipotential during androgen-mediated regeneration. (A) Time course for lineage-marking and androgen-mediated regeneration. (B) Percentage of single YFP$^+$ cells or YFP$^+$ clusters of 2 cells, 3–4 cells, and >4 cells at 4, 7, 14, and 28 days of androgen-mediated regeneration. This analysis does not include YFP$^+$AR$^+$ cells that fail to undergo AR deletion in the experimental mice; full quantitation of all cell populations is provided in *Figure 2—source data 1*. (C) YFP$^+$ cells (arrows) in prostates of mice with lineage-marked CARNs (top) and AR-deleted CARNs (bottom) at days 4, 7, 14 and 28 days during androgen-mediated regeneration. (D) Identification of basal YFP$^+$ cells (arrows) as progeny of CARNs (top) or AR-deleted CARNs (bottom). Scale bars in C) and D) correspond to 50 μm.

DOI: https://doi.org/10.7554/eLife.28768.005

The following source data is available for figure 2:

**Source data 1.** Quantitation of YFP$^+$ cells during regeneration.

DOI: https://doi.org/10.7554/eLife.28768.006

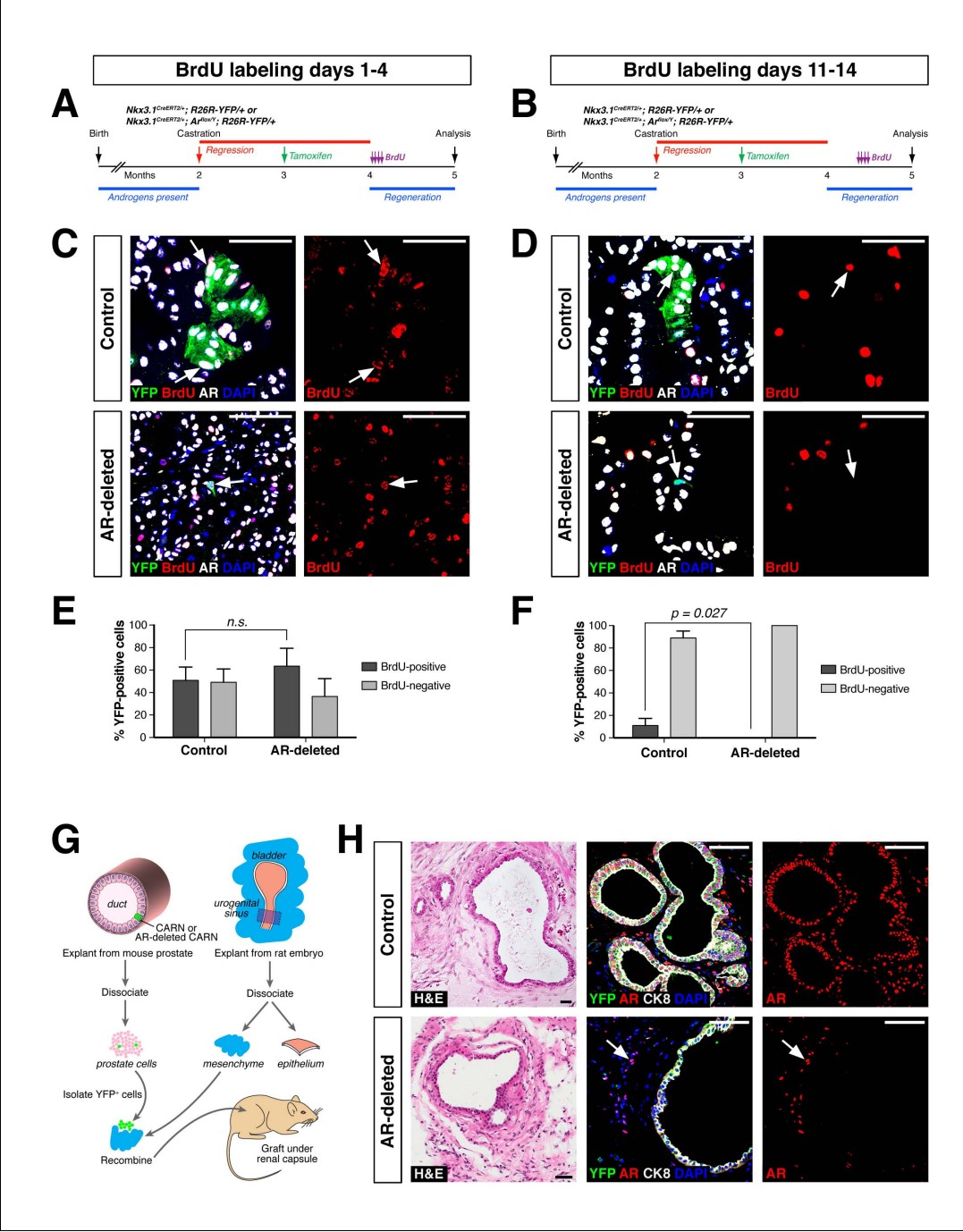

**Figure 3.** AR-deleted CARNs and/or their progeny have defects in proliferation during regeneration and in renal grafts. (A,B) Time course of BrdU incorporation during androgen-mediated regeneration of castrated and tamoxifen-treated control *Nkx3.1^CreERT2/+; R26R-YFP/+* mice and *Nkx3.1^CreERT2/+; Ar^flox/Y; R26R-YFP/+* mice. BrdU injections were performed during either days 1 through 4 (A) or days 11 through 14 (B), followed by analysis at 28 days. (C) Identification of BrdU^+YFP^+ cells (arrows) in control (top) and AR-deleted (bottom) prostate tissue after administration of BrdU during early stages of regeneration. (D) YFP-positive cells in control prostate tumors (top) can incorporate BrdU (arrow) but not in AR-deleted prostate tumors (bottom), after administration of BrdU during later stages of regeneration. (E,F) Percentage of BrdU^+ and BrdU^− cells among total YFP^+ cells after injection of BrdU from days 1 through 4 (E) or days 11 through 14 (F) of regeneration. Error bars represent one standard deviation; the difference in (E) is not statistically significant (p=0.34, independent t-test), but is significant in (F) (p=0.027, independent t-test). This analysis excludes YFP^+AR^+ cells that fail to undergo AR deletion in the experimental mice; full quantitation of all cell populations is provided in *Figure 3—source data 1*. (G) Schematic

*Figure 3 continued on next page*

*Figure 3 continued*

depiction of tissue recombination of lineage-marked CARNs with rat urogenital mesenchyme followed by renal grafting. (H) Analysis of grafts generated from lineage-marked CARNs (top) and AR-deleted CARNs (bottom); arrows in bottom panels indicate AR-expressing stromal cells surrounding the AR-negative prostate duct. Scale bars in C), D) and H) correspond to 50 μm.

DOI: https://doi.org/10.7554/eLife.28768.007

The following source data is available for figure 3:

**Source data 1.** Quantitation of BrdU incorporation and renal grafting data.

DOI: https://doi.org/10.7554/eLife.28768.008

significantly less efficient (12.5% graft efficiency, n = 16) compared to the control CARNs (p=0.003; 68.8% graft efficiency, n = 16), consistent with a proliferation defect in the AR-deleted CARNs.

Based on these findings, we further investigated the properties of CARNs and AR-deleted CARNs by establishing adherent cell lines. Using a novel method based on conditions that we previously established for culture of prostate organoids (*Chua et al., 2014*), we successfully generated adherent cell lines from single YFP$^+$ cells isolated from castrated and tamoxifen-treated *Nkx3.1$^{CreERT2/+}$; Ar$^{flox/Y}$; R26R-YFP/+* mice. Genotyping of the resulting lines led to identification of *Ar*-positive (non-recombined allele) and *Ar*-negative (recombined allele) lines, which we term APCA and ADCA (Ar-Positive CArn-derived and Ar-Deleted CArn-derived) lines. These cell lines could be propagated as adherent cells in the presence of Matrigel and DHT. Under these conditions, we found that the APCA (n = 2) and ADCA (n = 2) lines were morphologically indistinguishable (*Figure 4A*). These cell lines were comprised of a mixture of cells expressing basal (CK5) or luminal (CK8) markers or both, as well as Foxa1, an epithelial marker that encodes a transcriptional partner of AR (*Gao et al., 2003*; *He et al., 2010*) (*Figure 4A*). Furthermore, both the APCA and ADCA lines showed robust proliferation at similar levels, as demonstrated by Ki67 immunostaining, CellTiter-Glo assays, and colony formation in the presence or absence of DHT (*Figure 4A–C*).

To determine the relative efficiency of forming APCA and ADCA lines from AR$^+$ and AR$^-$ CARNs, respectively, we sorted 60 single YFP$^+$ cells from castrated and tamoxifen-treated *Nkx3.1$^{CreERT2/+}$; Ar$^{flox/Y}$; R26R-YFP/+* mice into individual wells of a 96-well plate. We found that six YFP$^+$ cells gave rise to adherent lines, with four of these corresponding to AR$^+$ lines that had failed to undergo Cre-mediated recombination of the conditional *Ar* allele, and two lines corresponding to AR$^-$ lines. After correcting for the 87.1% efficiency of recombination of the AR-floxed allele in vivo, these data indicate that the relative plating efficiency for the AR$^-$ CARNs compared to AR$^+$ CARNs is 7.4%, consistent with the decreased grafting efficiency of AR$^-$ CARNs.

Notably, we were also able to use this methodology to establish 14 primary human prostate epithelial cell lines from benign prostatectomy specimens at high efficiency. Similar to the mouse APCA cell lines, these HPE (Human Prostate Epithelial) cell lines are propagated as adherent cells in the presence of Matrigel and DHT. All these lines display similar marker phenotypes, expressing basal and luminal markers as well as AR and PSA, and are highly proliferative (*Figure 4—figure supplement 1*).

To assess the ability of the APCA and ADCA cell lines to reconstitute prostate ducts, we performed tissue recombination assays by combining 1 × 10$^5$ cells with rat urogenital mesenchyme followed by renal grafting. We found that the APCA lines could generate prostate ducts (n = 10 grafts with two lines; 100% efficiency), some with evidence of secretions, whereas the ADCA lines (n = 6 grafts with one line; 67% efficiency) generated ducts that lacked prostate secretions (*Figure 4D*). Next, we tested the role of AR in this tissue reconstitution assay by treating the mice grafted with APCA cells (n = 12 grafts with two lines) with tamoxifen at 7 weeks after grafting in order to induce *Ar* deletion. We found that tamoxifen treatment resulted in grafts containing prostate ducts composed of a mixture of AR-positive and negative cells, but with a decreased efficiency of graft formation relative to the same APCA lines in the absence of tamoxifen (42% versus 100% efficiency) (*Figure 4D*). Taken together, these results show that AR deletion decreases the efficiency of prostate duct formation by CARN-derived cells, consistent with the results obtained using AR-deleted CARNs (*Figure 3H*). Notably, since ADCA cells do not display a growth disadvantage relative to

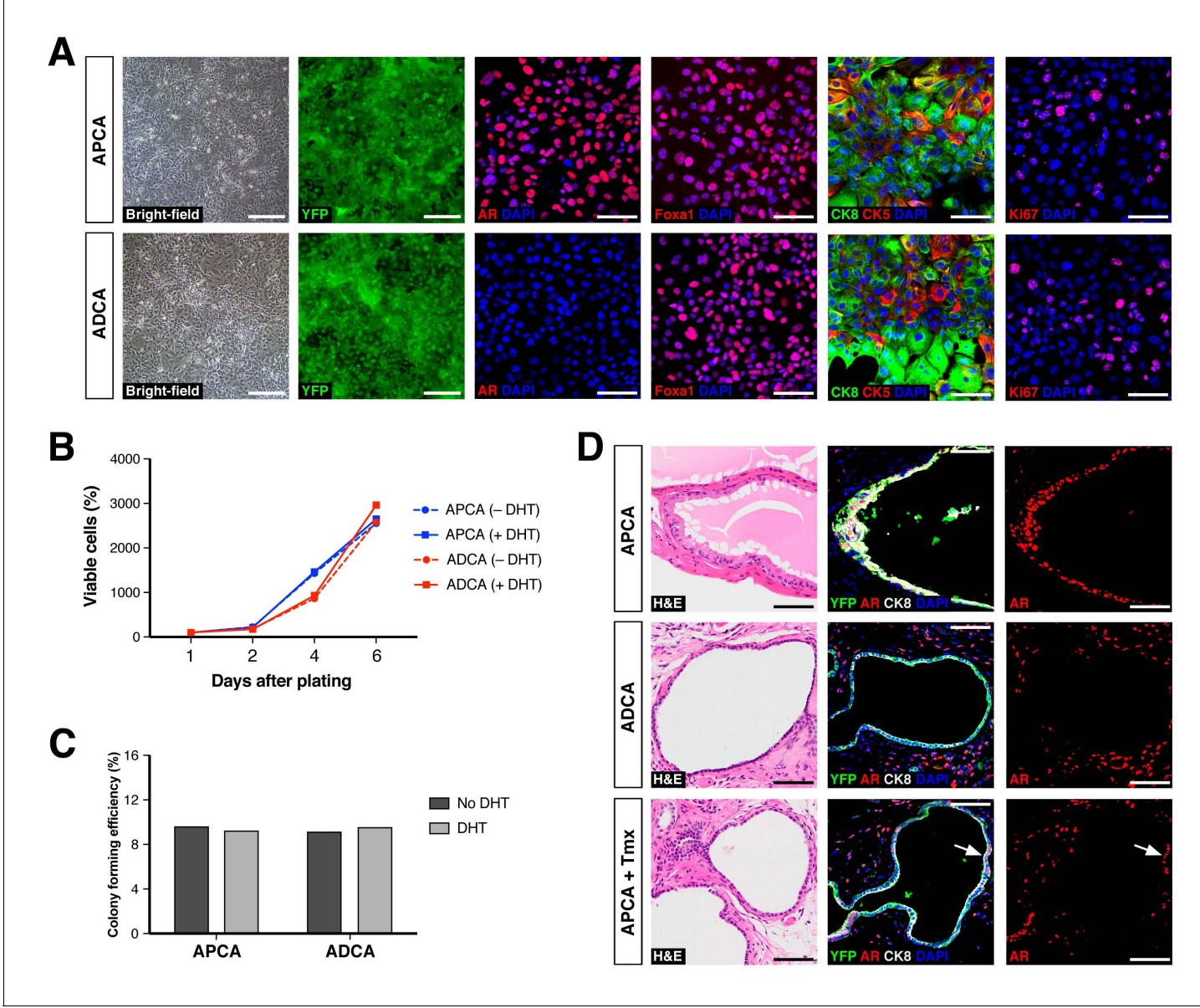

**Figure 4.** Properties of cell lines established from CARNs and AR-deleted CARNs. (A) Morphology and marker expression of cell lines derived from single YFP+ cells from castrated and tamoxifen-treated control *Nkx3.1*$^{CreERT2/+}$; *R26R-YFP/+* mice and *Nkx3.1*$^{CreERT2/+}$; *Ar*$^{flox/Y}$; *R26R-YFP/+* mice. The APCA lines (top) and ADCA lines (bottom) show similar bright-field morphology, expression of YFP, Foxa1, and Ki67, as well as co-expression of CK8 and CK5, but differ in expression of AR. (B) APCA and ADCA cell lines display similar cell growth at days 1, 2, 4, and 6 after plating in the absence or presence of DHT, as assessed by CellTiter-Glo assay. Results shown are from a single experiment with five technical replicates and are representative of two biological replicates after normalization with day 0 luminescent signal. (C) Colony formation by APCA and ADCA cell lines in the absence or presence of DHT. Results are from a single experiment with three technical replicates and are representative of two biological replicates. (D) Renal grafts generated from tissue recombinants of 100,000 APCA or ADCA cells with rat urogenital mesenchyme, and analyzed at 12 weeks. Bottom row shows APCA grafts treated with tamoxifen for 4 days at 7 weeks of growth to induce *Ar* deletion (bottom); arrows indicate cells that did not undergo *Ar* deletion after tamoxifen treatment. Scale bars in A) and D) correspond to 50 μm.

DOI: https://doi.org/10.7554/eLife.28768.009

The following source data and figure supplement are available for figure 4:

**Source data 1.** Epithelial cell lines established from mouse and human prostate tissue.
DOI: https://doi.org/10.7554/eLife.28768.011
**Figure supplement 1.** Establishment of novel human prostate epithelial cell lines.
DOI: https://doi.org/10.7554/eLife.28768.010

APCA cells in culture, this difference in duct formation is likely to be due to a non-cell-autonomous effect mediated by the urogenital mesenchyme in grafts.

To examine the molecular basis for differences between the ADCA and APCA lines (n = 2 lines each), we performed RNA-sequencing followed by bioinformatic analyses. Based on the RNA expression profiling data, we constructed a differential expression signature comparing ADCA cells to APCA cells. Using the resulting ADCA signature to examine pathway enrichment by Gene Set Enrichment Analysis (GSEA) (*Subramanian et al., 2005*), we found up-regulation of gene sets involved in DNA replication and repair pathways, as well as cell cycle and apoptosis (*Figure 5A*), suggesting that cellular proliferation and survival are affected by AR deletion. We also compared the ADCA signature with a signature defined between expression profiles of AR-null and AR-positive mouse prostate luminal cells (*Xie et al., 2017*) and found enrichment for up-regulated genes (*Figure 5B*). Next, we performed a cross-species comparison of the ADCA signature with a signature defined between profiles of human prostate luminal and basal epithelial cells (*Zhang et al., 2016b*) and found that there was no significant enrichment in either tail (*Figure 5C*), indicating that AR deletion does not drive APCA cells towards a specific lineage. Furthermore, we performed GSEA comparisons of the ADCA signature with several signatures obtained from analyses of human prostate cancer progression. In particular, we observed enrichment for up-regulated genes when compared to a signature of CRPC from Best and colleagues (*Best et al., 2005*), as well as to a signature of metastatic CRPC from Stanbrough and colleagues (*Stanbrough et al., 2006*) (*Figure 5D,E*). Moreover, we observed a strong enrichment when compared to a signature from Beltran and colleagues (*Beltran et al., 2016*) defined between CRPC with neuroendocrine differentiation (CRPC-NE) and

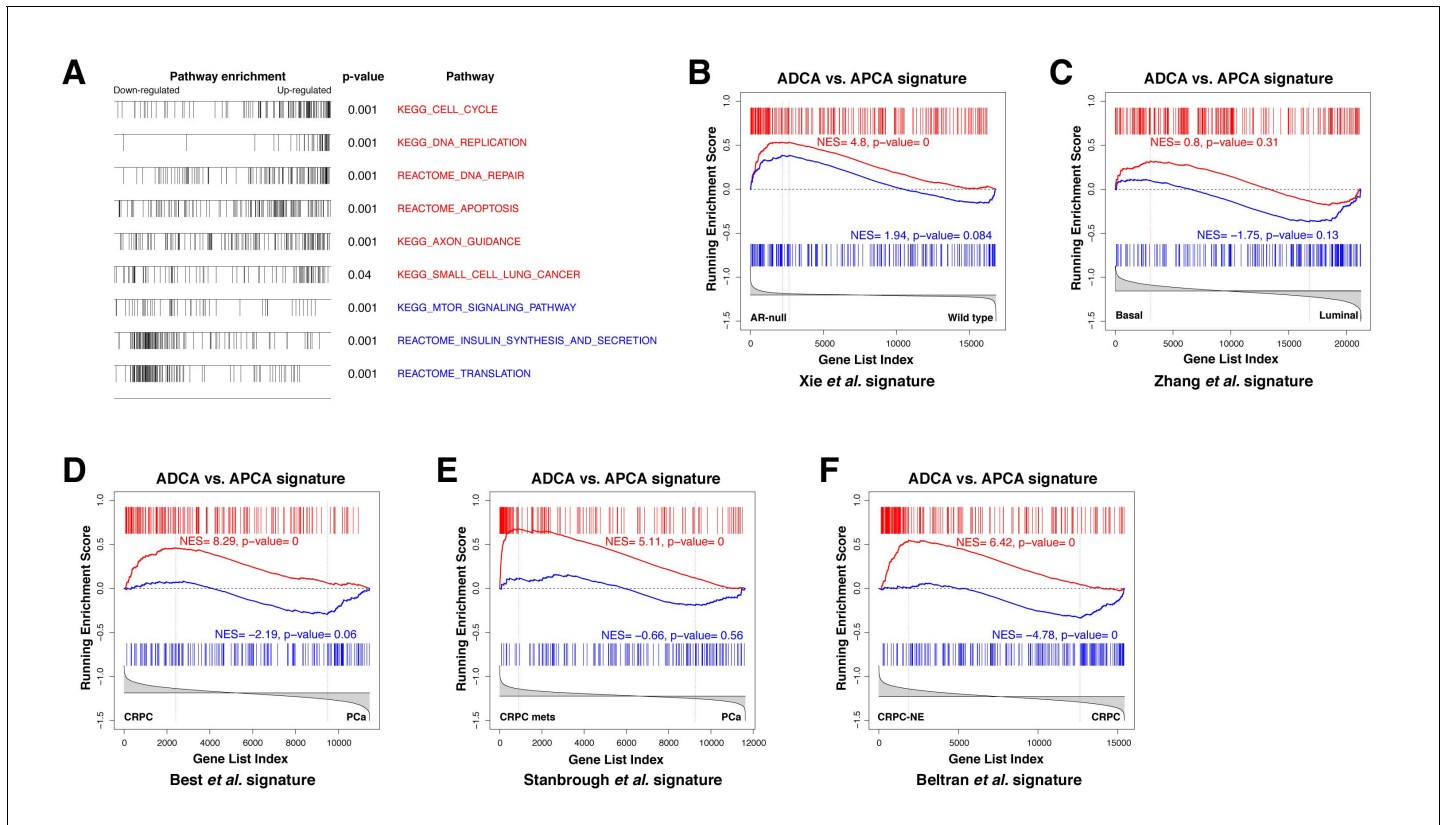

**Figure 5.** Gene set enrichment analysis of the ADCA signature. (**A**) Selected biological pathways that are enriched in the ADCA versus APCA signature. (**B**) GSEA plot showing enrichment in the positive tail for a signature of AR-null mouse prostate epithelial cells. (**C**) Cross-species GSEA showing lack of enrichment with a signature based on isolated human prostate basal and luminal epithelial populations. (**D–F**) Cross-species GSEA comparing the ADCA expression signature with three independent expression signatures based on tumor samples from human patients. NES: normalized enrichment score; p-value is calculated using 1000 gene permutations.

DOI: https://doi.org/10.7554/eLife.28768.012

non-neuroendocrine CRPC (*Figure 5F*), consistent with our observation of pathway enrichment for gene sets corresponding to axon guidance and small-cell lung cancer (*Figure 5A*).

Finally, we tested the ability of AR-deleted CARNs to serve as a cell of origin for prostate cancer, based on the previous finding that prostate cancer can initiate from CARNs after specific deletion of *Pten* and androgen-mediated regeneration (*Wang et al., 2009*). We used $Nkx3.1^{CreERT2/+}$; $Pten^{flox/flox}$; *R26R-YFP/+* controls (which we term NP-CARN) and $Nkx3.1^{CreERT2/+}$; $Pten^{flox/flox}$; $Ar^{flox/Y}$; *R26R-YFP/+* mice (NPA-CARN) in an experimental paradigm involving castration, tamoxifen-treatment, and androgen-mediated regeneration for one month. We found that AR deletion resulted in a significant difference between the NP-CARN and NPA-CARN phenotypes, as the NP-CARN controls displayed high-grade prostatic intraepithelial neoplasia (PIN), whereas the NPA-CARN prostates showed a weak phenotype corresponding to diffuse hyperplasia with mild inflammation and increased apoptosis, (*Figure 6A*). The NPA-CARN prostates contained YFP-positive cells that also expressed phospho-Akt (pAkt), indicating successful deletion of *Pten,* but these cells were only found as solitary or as small clusters, unlike the large clusters of YFP⁺pAkt⁺ cells observed in the control NP prostates (*Figure 6A*). Furthermore, the NPA-CARN prostates displayed a decreased proliferative index relative to NP-CARN (2.7%, n = 3 vs. 9.2%, n = 3), as well as increased apoptosis (2.6%, n = 3 vs. 0.7%, n = 3) (*Figure 6B*). Taken together, findings indicate that AR is required for tumor initiation following *Pten* deletion in CARNs.

In contrast, AR deletion did not affect tumor initiation following combined deletion of *Pten* and activation of the oncogenic $Kras^{G12D}$ allele. Using a similar protocol for castration, tamoxifen-treatment, and androgen-mediated regeneration, we compared the phenotypes of $Nkx3.1^{CreERT2/+}$; $Pten^{flox/flox}$; $Kras^{LSL-G12D/+}$; *R26R-YFP/+* controls (NPK-CARN) and $Nkx3.1^{CreERT2/+}$; $Pten^{flox/flox}$; $Kras^{LSL-G12D/+}$; $Ar^{flox/Y}$; *R26R-YFP/+* mice (NPKA-CARN). In both genotypes, deletion of *Pten* and activation of oncogenic *Kras* resulted in formation of tumors with large clusters of YFP⁺ cells that express pAkt and Ras (*Figure 6C*). Furthermore, both NPK-CARN and NPKA-CARN tumors displayed high proliferative indices (20%, n = 3 vs. 19%, n = 3) and low frequencies of apoptosis (0.9%, n = 3 vs. 0.8%, n = 3) (*Figure 6D*). Notably, we observed an important difference between the NPK-CARN and NPKA-CARN tumors, as all the NPKA-CARN tumors contained a low but variable percentage of synaptophysin-positive neuroendocrine cells among total epithelial cells (0.7%, n = 3), which were never observed in the NPK-CARN controls (0%, n = 3) (*Figure 6E,F*). We also observed rare cells in all three NPKA-CARN tumors that expressed other neuroendocrine markers such as Chromogranin A, Foxa2, and Aurora kinase A (*Figure 6E*). Since the synaptophysin-postive cells co-expressed YFP (*Figure 6E*), we conclude that transformed AR-negative CARNs can give rise to neuroendocrine cells.

## Discussion

Taken together, our analyses have defined specific roles for AR in regulating the progenitor properties of CARNs, and indicate that the intrinsic castration-resistance of CARNs is independent of AR function. We find that targeted deletion of AR does not affect the percentage of CARNs, their luminal marker expression, or their ability to generate basal cells during androgen-mediated regeneration. However, there are fewer luminal progeny from AR-deleted CARNs during regeneration in vivo, and there is a decreased efficiency of prostate duct formation by both AR-deleted CARNs and ADCA cells in renal grafts. Thus, AR deletion in CARNs may primarily affect the proliferation and/or survival of their luminal progeny in vivo, although an effect on CARNs themselves cannot be excluded.

Interestingly, our results suggest potential roles of the stroma in modulating the proliferation of CARNs and/or their luminal progeny. Notably, BrdU incorporation assays reveal a proliferation defect of AR-deleted CARNs during later stages of regeneration but not during early regeneration. One possible explanation is that AR activity may cell-autonomously regulate the proliferation of luminal progeny of CARNs; alternatively, however, stromal remodeling during later stages of regeneration may alter non-cell autonomous signals that regulate luminal proliferation. Furthermore, since ADCA cells do not display a growth defect in culture, their decreased efficiency of prostate duct formation in grafts is likely due to a non-cell-autonomous inhibitory effect from the stroma.

Our study has also yielded interesting insights into differences between CARNs and other luminal epithelial cells. While this manuscript was in preparation, another study also investigated the

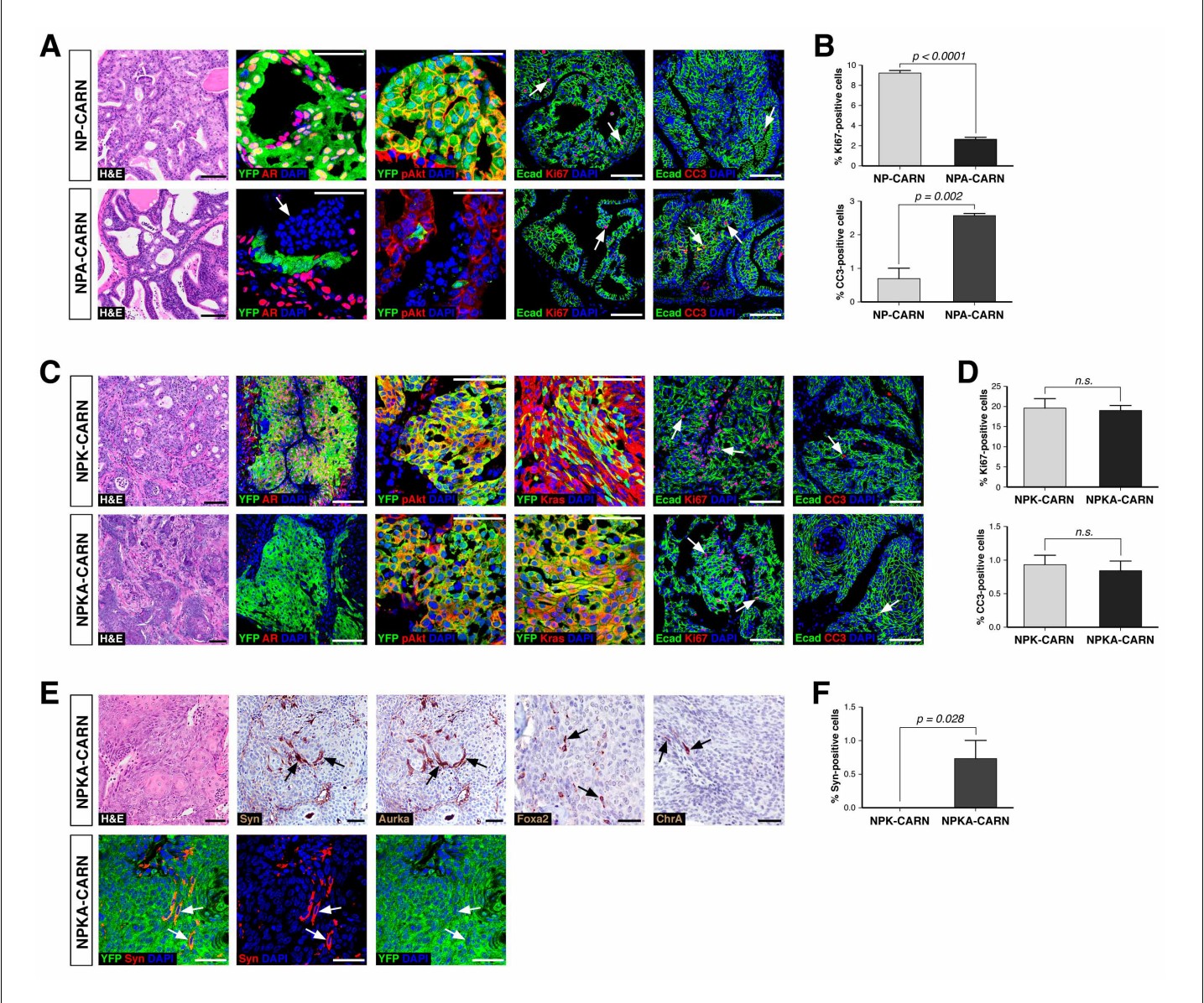

**Figure 6.** Deletion of AR alters the ability of CARNs to serve as a cell of origin for prostate cancer. (**A**) Prostate histology and marker expression in *Nkx3.1$^{CreERT2/+}$*; *Pten$^{flox/flox}$*; *R26R-YFP/+* (NP-CARN) and *Nkx3.1$^{CreERT2/+}$*; *Pten$^{flox/flox}$*; *Ar$^{flox/Y}$*; *R26R-YFP/+* (NPA-CARN) mice that have been castrated and tamoxifen-treated, followed by androgen-mediated regeneration for 1 month. Shown are representative images for hematoxylin-eosin staining (H and E) and immunofluorescence for YFP, AR, phospho-Akt (pAkt), E-cadherin (Ecad), Ki67, and cleaved caspase-3 (CC3). Arrows indicate occurrence of cell death (YFP/AR in NPA-CARN), proliferation (Ecad/Ki67), and apoptosis (Ecad/CC3). (**B**) Quantitation of Ki67$^+$ and CC3$^+$-positive cells in total Ecad$^+$ epithelial cells in NP-CARN and NPA-CARN prostates. Error bars represent one standard deviation; differences between groups are statistically significant as determined by independent t-test. (**C**) Prostate tumor histology and marker expression in *Nkx3.1$^{CreERT2/+}$*; *Pten$^{flox/flox}$*; *Kras$^{LSL-G12D/+}$*; *R26R-YFP/+* (NPK-CARN) and *Nkx3.1$^{CreERT2/+}$*; *Pten$^{flox/flox}$*; *Kras$^{LSL-G12D/+}$*; *Ar$^{flox/Y}$*; *R26R-YFP/+* (NPKA-CARN) mice that have been castrated and tamoxifen-treated, followed by androgen-mediated regeneration for 1 month. Arrows indicate cells undergoing proliferation (Ecad/Ki67) and apoptosis (Ecad/CC3). (**D**) Quantitation of Ki67$^+$ and CC3$^+$-positive cells in total Ecad$^+$ epithelial cells in NPK-CARN and NPKA-CARN prostates. Differences between groups are not statistically significant as determined by independent t-test (Ki67, p=0.724; CC3, p=0.507). (**E**) Focal neuroendocrine differentiation in NPKA-CARN tumors. Shown are H and E and immunohistochemical staining (IHC) of serial sections for Synaptophysin (Syn) and Aurora kinase A (Aurka), IHC for Foxa2 and Chromogranin A (ChrA), as well as immunofluorescence for YFP and Syn shown as an overlay and as individual channels; arrows indicate positive cells. (**F**) Quantitation of Syn$^+$ cells in total epithelial cells in NPK-CARN and NPKA-CARN tumors. Scale bars for H and E and IHC in **A, C,**) and **E**) correspond to 100 μm, and in other panels to 50 μm.

DOI: https://doi.org/10.7554/eLife.28768.013

The following source data is available for figure 6:

*Figure 6 continued on next page*

*Figure 6 continued*

**Source data 1.** Tumor phenotypes and marker quantitation.

DOI: https://doi.org/10.7554/eLife.28768.014

requirements of AR in CARNs, and reported that AR-deleted CARNs completely failed to generate progeny during regeneration (*Xie et al., 2017*). This apparent discrepancy may be partially explained by our observation that AR-deleted CARNs can still generate basal progeny, and by the failure of progeny from AR-deleted CARNs to proliferate at later stages of androgen-mediated regeneration. However, we concur that CARNs require AR function to generate viable luminal progeny, which is not the case for most luminal cells during homeostasis or regeneration (*Xie et al., 2017*; *Zhang et al., 2016a*). Furthermore, the decreased proliferation of AR-deleted CARNs during regeneration contrasts with the transient increase in luminal proliferation observed after inducible AR deletion in the adult prostate epithelium, which is also a non-cell-autonomous effect mediated by the stroma (*Zhang et al., 2016a*). Together with our previous finding that CARNs display increased organoid formation efficiency relative to other luminal cells (*Chua et al., 2014*), these findings support the identification of CARNs as a distinct luminal population with stem/progenitor properties, and highlight the complexity of AR functions in the epithelial and stromal compartments.

In addition, we note that Xie and colleagues reported that *Pten* deletion in CARNs resulted in tumor formation after regeneration (*Xie et al., 2017*), unlike the absence of tumors that we observe in NPA-CARN mice. At present, the basis for this discrepancy remains unclear. Our finding that AR deletion results in failure of tumor formation following *Pten* inactivation could be due to differences between CARNs and bulk luminal cells and/or to differences due to *Pten* loss in the regressed versus hormonally intact prostate epithelium. In principle, these possibilities could potentially be distinguished using inducible Cre-drivers to delete *Pten* in bulk luminal cells in regressed versus hormonally intact mice.

Our findings on CARNs in mouse models may be of potential relevance for human prostate biology and cancer. Although CARNs are defined in the regressed prostate epithelium, and our in vivo studies involve manipulations performed after castration in mice, there is evidence that CARN-like cells exist in the human prostate from tissue-slice culture experiments (*Zhao et al., 2010*), as well as from analyses of prostate tumors after androgen-deprivation (*Germann et al., 2012*). However, it is less clear whether multipotent luminal progenitors can be identified in the context of the hormonally intact human prostate. Previous lineage-reconstruction studies using patterns of mitochondrial DNA mutations have indicated the existence of multipotent epithelial progenitors (*Blackwood et al., 2011*; *Gaisa et al., 2011*), and recent work has provided evidence for multipotent basal progenitors localized to the most proximal region of the prostate as well as more distally located unipotent luminal progenitors (*Moad et al., 2017*). Notably, *ex vivo* studies of human prostate organoids have demonstrated the existence of bipotential luminal progenitors (*Karthaus et al., 2014*). Thus, we believe that current data favor a general similarity of epithelial lineage relationships in the two species, suggesting that findings deduced from analyses of mice may be translatable to the human prostate.

The ability of CARNs to retain progenitor properties even in the absence of AR raises the possibility that CARNs represent a cell of origin for prostate cancers that are particularly susceptible to develop castration-resistance. Notably, under conditions of AR down-regulation, such as those that may occur during aging or inflammation, CARNs that lack tumor suppressors such as PTEN may represent a latent target for subsequent oncogenic events that can confer tumor growth, such as those activating the ERK MAP kinase pathway. Interestingly, our bioinformatic analyses of the ADCA cell line signature shows enrichment with castration-resistance signatures based on expression data from human prostate cancer patients (Best and Stanbrough signatures), consistent with increasing evidence supporting AR-independent mechanisms of castration-resistance (*Beltran et al., 2014*; *Vlachostergios et al., 2017*; *Watson et al., 2015*). In addition, the observed enrichment with the Beltran CRPC-NE signature suggests a similarity in gene expression programs with advanced cancers that lack AR activity, as neuroendocrine differentiation in prostate tumors is associated with loss of AR expression (*Beltran et al., 2011*). Notably, consistent with a role for AR loss in the emergence of neuroendocrine phenotypes, tumors in NPKA-CARN mice can display focal neuroendocrine

differentiation, which has also been recently described in other mouse models of advanced prostate cancer (**Ku et al., 2017**; **Zou et al., 2017**).

In this regard, we note that the NP-CARN and NPK-CARN mice develop tumor phenotypes similar to those in NP and NPK mice, which have the same genotypes but whose tumors are induced by the *Nkx3.1^{CreERT2}* driver in hormonally intact adult prostate (**Aytes et al., 2013**; **Floc'h et al., 2012**). Interestingly, NP tumors are initially castration-sensitive (**Floc'h et al., 2012**), consistent with the inability of NPA-CARN mice to develop tumors, whereas NPK tumors are castration-resistant (**Aytes et al., 2013**), consistent with the phenotype of NPKA-CARN tumors. The molecular basis for this switch is currently unclear, but it is conceivable that it involves ETS family transcription factors, which are known to interact with AR to positively and negatively modulate its activity (**Baena et al., 2013**; **Bose et al., 2017**; **Chen et al., 2013**); interestingly, ETV4 is up-regulated in NPK tumors and may be involved in this switch (**Aytes et al., 2013**). However, the focal neuroendocrine differentiation observed in NPKA-CARN tumors suggests that oncogenic transformation of AR-deleted CARNs can also facilitate transdifferentiation of luminal cells to neuroendocrine fates, as we have demonstrated for a *Pten* and *Trp53* mutant mouse model (NPp53) after anti-androgen treatment (**Zou et al., 2017**).

Finally, since tumors initiated from CARNs following combined *Pten* deletion and *Kras* activation are at least partially independent of AR from their outset, it is conceivable that such tumors are intrinsically more resistant to second-generation anti-androgen therapies. Interestingly, recent studies have also identified distinct castration-resistant progenitors that express Bmi1 (CARBs) that are cells of origin for prostate cancer (**Yoo et al., 2016**). The development of targeted therapies directed at molecular features of CARNs and/or other castration-resistant luminal cells may therefore be relevant for successful combination with anti-androgen therapies.

# Materials and methods

## Key resources table

| Reagent type (species) or resource | Designation | Source or reference | Identifiers | Additional information |
|---|---|---|---|---|
| Strain (*M. musculus*) | NOG | PMID: 15879151 | NOD.Cg-*Prkdc^{scid} Il2rg^{tm1Sug}*/JicTac | Taconic (Hudson, NY) |
| Strain (*M. musculus*) | *Nkx3.1^{CreERT2}* | PMID: 19741607 | *Nkx3-1^{tm4(CreERT2)Mms}* | established by Shen lab |
| Strain (*M. musculus*) | *Pten^{flox}* | PMID: 11691952 | C;129S4-*Pten^{tm1Hwu}*/J | JAX #004597 (Bar Harbor, ME) |
| Strain (*M. musculus*) | *Kras^{LSL-G12D}* | PMID: 11751630 | B6.129-*Kras^{tm4Tyj}*/Nci | MMHCC #01XJ6 |
| Strain (*M. musculus*) | *AR^{flox}* | PMID: 14745012 | B6N.129-*Ar^{tm1Verh}*/Cnrm | EMMA #02579 |
| Strain (*M. musculus*) | *R26R-YFP* | PMID: 11299042 | B6.129 × 1-*Gt(ROSA) 26Sor^{tm1(EYFP)Cos}*/J | JAX #006148 |
| Cell line (*Homo sapiens*) | HPE-1 | this work | | Adherent cell line established from radical prostatectomy 23 tissue, sorted for EpCAM⁺Ecad⁺ cells |
| Cell line (*H. sapiens*) | HPE-2 | this work | | Adherent cell line established from radical prostatectomy 23 tissue, sorted for EpCAM⁺Ecad⁺ cells |
| Cell line (*H. sapiens*) | HPE-3 | this work | | Adherent cell line established from radical prostatectomy 24 tissue, sorted for EpCAM⁺Ecad⁺ cells |
| Cell line (*H. sapiens*) | HPE-4 | this work | | Adherent cell line established from radical prostatectomy 25 tissue |
| Cell line (*H. sapiens*) | HPE-5 | this work | | Adherent cell line established from radical prostatectomy 25 tissue, sorted for EpCAM⁺Ecad⁺ cells |
| Cell line (*H. sapiens*) | HPE-6 | this work | | Adherent cell line established from radical prostatectomy 25 tissue, sorted for EpCAM⁺Ecad⁺Ngfr⁺ cells |

*Continued on next page*

*Continued*

| Reagent type (species) or resource | Designation | Source or reference | Identifiers | Additional information |
|---|---|---|---|---|
| Cell line (H. sapiens) | HPE-7 | this work | | Adherent cell line established from radical prostatectomy 25 tissue, sorted for EpCAM$^+$Ecad$^+$Cd24$^+$ cells |
| Cell line (H. sapiens) | HPE-8 | this work | | Adherent cell line established from radical prostatectomy 26 tissue |
| Cell line (H. sapiens) | HPE-9 | this work | | Adherent cell line established from radical prostatectomy 26 tissue, sorted for EpCAM$^+$Ecad$^+$ cells |
| Cell line (H. sapiens) | HPE-10 | this work | | Adherent cell line established from radical prostatectomy 26 tissue, sorted for EpCAM$^+$Ecad$^+$Cd24$^+$ cells |
| Cell line (H. sapiens) | HPE-11 | this work | | Adherent cell line established from radical prostatectomy 26 tissue, sorted for EpCAM$^+$Ecad$^+$Agr2$^+$ cells |
| Cell line (H. sapiens) | HPE-12 | this work | | Adherent cell line established from radical prostatectomy 27 tissue |
| Cell line (H. sapiens) | HPE-13 | this work | | Adherent cell line established from radical prostatectomy 27 tissue; sorted for EpCAM$^+$Ecad$^+$ cells |
| Cell line (H. sapiens) | HPE-14 | this work | | Adherent cell line established from radical prostatectomy 27 tissue, sorted for EpCAM$^+$Ecad$^+$Cd24$^+$ cells |
| Cell line (M. musculus) | ADCA-1 | this work | | Adherent cell line established from single YFP$^+$ cell isolated from castrated and tamoxifen-treated Nkx3.1$^{CreERT2/+}$; Ar$^{flox/Y}$; R262R-YFP/+ mouse with deleted Ar (recombined) allele |
| Cell line (M. musculus) | ADCA-2 | this work | | Adherent cell line established from single YFP$^+$ cell isolated from castrated and tamoxifen-treated Nkx3.1$^{CreERT2/+}$; Ar$^{flox/Y}$; R262R-YFP/+ mouse with deleted Ar (recombined) allele |
| Cell line (M. musculus) | APCA-1 | this work | | Adherent cell line established from single YFP$^+$ cell isolated from castrated and tamoxifen-treated Nkx3.1$^{CreERT2/+}$; Ar$^{flox/Y}$; R262R-YFP/+ mouse with intact Ar (non-recombined) allele |
| Cell line (M. musculus) | APCA-2 | this work | | Adherent cell line established from single YFP$^+$ cell isolated from castrated and tamoxifen-treated Nkx3.1$^{CreERT2/+}$; Ar$^{flox/Y}$; R262R-YFP/+ mouse with intact Ar (non-recombined) allele |
| Antibody | Androgen receptor (AR) | Sigma (St. Louis, MO) | A9853 | |
| Antibody | Cytokeratin 8 (CK8) | Developmental Studies Hybridoma Bank (Iowa City, IA) | TROMA-1 | |
| Antibody | Cytokeratin 18 (CK18) | Abcam (Cambridge, MA) | ab668 | |

*Continued on next page*

Continued

| Reagent type (species) or resource | Designation | Source or reference | Identifiers | Additional information |
|---|---|---|---|---|
| Antibody | Cytokeratin 5 (CK5) | Covance (San Diego, CA) | SIG3475 | |
| Antibody | Cytokeratin 5 (CK5) | Covance | PRB-160P | |
| Antibody | p63 | Santa Cruz (Dallas, TX) | sc-8431 | |
| Antibody | GFP | Abcam | ab13970 | |
| Antibody | GFP | Roche (St. Louis, MO) | 11814460001 | |
| Antibody | BrdU | AbD Serotec MCA (Hercules, CA) | 2060 | |
| Antibody | Foxa1 | Abcam | ab55178 | |
| Antibody | Ki67 | eBiosciences (San Diego, CA) | 14–5698, clone SolA15 | |
| Antibody | Cleaved-caspase-3 (CC3) | BD Pharmingen (San Jose, CA) | 559565 | |
| Antibody | Prostate specific antigen (PSA) | Dako (Santa Clara, CA) | M0750, clone ER-PR8 | |
| Antibody | Kras | Abcam | ab84573 | |
| Antibody | Synaptophysin (Syn) | BD Transduction Laboratories (San Jose, CA) | 611880 | |
| Antibody | Aurora A (Aurka) | Abcam | ab13824 | |
| Antibody | Chromogranin A (ChrA) | Abcam | ab15160 | |
| Antibody | Foxa2 | Abnova (Taiwan) | H00003170-M12 | |
| Antibody | AMACR | Zeta Corp (Arcadia, CA) | Z2001 | |
| Antibody | EpCAM | BioLegend | 118214 | |
| Antibody | E-cadherin | eBiosciences | 46-3249-82 | |
| Antibody | Nerve growth factor receptor (Ngfr) | BioLegend | 345108 | |
| Antibody | Cd24 | BioLegend | 311008 | |
| Antibody | Anterior gradient 2 (Agr2) | Abcam | ab1139894 | |
| Antibody | EpCAM | BioLegend | 324208 | |
| Sequence-based reagent | $Nkx3.1$ wild-type primers | PMID: 19741607 | DOI 10.1038/nature 08361 | |
| Sequence-based reagent | $Nkx3.1^{CreERT2}$ primers | PMID: 19741607 | DOI 10.1038/nature 08361 | |
| Sequence-based reagent | $CreER^{T2}$ primers | PMID: 19741607 | DOI 10.1038/nature 08361 | |
| Sequence-based reagent | $R262R$-$YFP$ primers | PMID: 11299042 | | |
| Sequence-based reagent | $Pten^{flox}$ primers | PMID: 11691952 | DOI: 10.1126/science. 1065518 | |
| Sequence-based reagent | $Pten$ wild-type primers | PMID: 11691952 | DOI: 10.1126/science. 1065518 | |
| Sequence-based reagent | $Kras^{LSL-G12D}$ primers | PMID: 11751630 | DOI: 10.1101/gad. 943001 | |
| Sequence-based reagent | $Kras$ wild-type primers | PMID: 11751630 | DOI: 10.1101/gad. 943001 | |

*Continued*

| Reagent type (species) or resource | Designation | Source or reference | Identifiers | Additional information |
|---|---|---|---|---|
| Sequence-based reagent | $Ar^{flox}$ primers | PMID: 14676301 | DOI: 10.1084/jem.20031233 | |
| Sequence-based reagent | Ar wild-type primers | PMID: 14676301 | DOI: 10.1084/jem.20031233 | |
| Sequence-based reagent | $Ar^{flox}$ (recombined) primers | PMID: 14676301 | DOI: 10.1084/jem.20031233 | |
| Sequence-based reagent | $Ar^{flox}$ (not recombined) primers | PMID: 14676301 | DOI: 10.1084/jem.20031233 | |
| Commercial assay or kit | Tyramide amplification | ThermoFisher Scientific (Waltham, MA) | T20922 | |
| Ccommercial assay or kit | Tyramide amplification | ThermoFisher Scientific | T30953 | |
| Commercial assay or kit | Tyramide amplification | ThermoFisher Scientific | T30954 | |
| Commercial assay or kit | Tyramide amplification | ThermoFisher Scientific | T20926 | |
| Commercial assay or kit | Tyramide amplification | ThermoFisher Scientific | T20912 | |
| Commercial assay or kit | ABC Elite | Vector Labs (Burlingame, CA) | pk6101 | |
| Commercial assay or kit | Citrate-based antigen unmasking solution | Vector Labs | H3300 | |
| Commercial assay or kit | Tris-based antigen unmasking solution | Vector Labs | H3301 | |
| Commercial assay or kit | NovaRED | Vector Labs | SK3800 | |
| Commercial assay or kit | CellTiter-Glo 3D | Promega (Madison, Wi) | G9681 | |
| Commercial assay or kit | MagMAX−96forMicroarrays Total RNA Isolation Kit | Ambion (Waltham, MA) | Am1839 | Used the 'no spin' protocol for RNA purification |
| Commercial assay or kit | TruSeq Stranded mRNA library prep kit | Illumina (San Diego, CA) | 20020595 | Library preparation was performed by the Columbia Genome Center using Illumina kits |
| Chemical compound, drug | Tissue Tek OCT compound | VWR Scientific (Radnor, PA) | 25608–930 | |
| Chemical compound, drug | Glutamax | Invitrogen (Waltham, MA) | 35050061 | |
| Chemical compound, drug | Tamoxifen; TM | Sigma | T5648-5G | |
| Chemical compound, drug | Gentamicin | Invitrogen | 15750–060 | |
| Chemical compound, drug | Collagenase/hyaluronidase | STEMCELL Technologies (Cambridge, MA) | 07912 | |
| Chemical compound, drug | Modified Hank's Balanced Salt Solution; HBSS | STEMCELL Technologies | 37150 | |
| Chemical compound, drug | Dnase I | STEMCELL Technologies | 07900 | |
| Chemical compound, drug | Y-27632 ROCK inhibitor | STEMCELL Technologies | 72307 | |
| Chemical compound, drug | 10x Earle's Balanced Salt Solution | ThermoFisher Scientific | 14155063 | |

*Continued*

| Reagent type (species) or resource | Designation | Source or reference | Identifiers | Additional information |
|---|---|---|---|---|
| Chemical compound, drug | Hepatocyte medium supplemented with epidermal growth factor (EGF) | Corning (Corning, NY) | 355056 | |
| Chemical compound, drug | Matrigel | ThermoFisher Scientific | 354234 | |
| Chemical compound, drug | 0.25% trypsin-EDTA | STEMCELL Technologies | 07901 | |
| Chemical compound, drug | FBS | ThermoFisher Scientific | 12676029 | |
| Chemical compound, drug | DMEM/F12 | ThermoFisher Scientific | 11320033 | |
| Chemical compound, drug | BrdU | Sigma | B5002 | |
| Chemical compound, drug | Dispase | STEMCELL Technologies | 07913 | |
| Chemical compound, drug | Dihydrotestosterone; DHT | Sigma | A8380 | |
| Software, algorithm | Real time analysis; RTA | Illumina | https://support.illumina.com/sequencing/sequencing_software/real-time_analysis_rta.html | Base calling using this software was performed by the Columbia Genome Center |
| Software, algorithm | bcl2fastq2 | Illumina | Ilumina: version 2.17 | The sequencing data was trimmed and converted to fastq format by the Columbia Genome Center |
| Software, algorithm | Spliced Transcripts Alignment to a Reference (STAR) | PMID: 23104886 | Github: version 2.5.2b | Sequencing reads mapping to mouse genome (USCS/mm10) was performed by the Columbia Genome Center |
| Software, algorithm | FeatureCounts | PMID: 24227677 | subread.sourceforge.net version: v1.5.0-p3 | Sequencing reads mapping to mouse genome (USCS/mm10) was performed by the Columbia Genome Center |
| Software, algorithm | R-studio 0.99.902, R v3.3.0 | The R Foundation for Statistical Computing, ISBN 3-900051-07-0 | v3.3.0 | R language for statistical computing was used for data analysis and visualization |
| Software, algorithm | homoloGene | NCBI | | |
| Software, algorithm | Gene Set Enrichment Analysis | PMID: 16199517 | DOI 10.1073/pnas.0506580102 | GSEA was used to compares differential gene expression signatures |
| Software, algorithm | Statistical Package for the Social Sciences; SPSS, Kolmogorov-Smirnov test, Arcsine transformation, Welch t-test, Fisher's Exact Test | IBM SPSS Statistics | | |
| Software, algorithm | Histological grading of mouse prostate phenotypes | PMID: 12163397 | DOI 10.1016/S0002-9440(10)64228-9 | |
| Other | Mini-osmotic pump | Alzet (Cupertino, CA) | 0000298 | |
| Other | 40 μm cell strainer | Falcon (Corning, NY) | Fisher Scientific 352340 | |
| Other | 96-well Primaria plate | Corning | Fisher Scientific 353872 | |
| Other | 6-well Primaria plate | Corning | Fisher Scientific 353846 | |
| Other | 96-well CELLSTAR plate | Greiner Bio-One (Monroe, NC) | 655090 | |
| Other | Lab-Tek Chamber Slide | Thermo Fisher Scientific | 154534 | |

## Mouse strains and genotyping

The *Nkx3.1^CreERT2* driver *(Nkx3-1^tm4(cre/ERT2)Mms)* has been previously described (*Wang et al., 2009*). Mice carrying the *R26R-YFP* (*B6.129* × *1-Gt(ROSA)26Sor^tm1(EYFP)Cos*/J) reporter (*Srinivas et al., 2001*) were obtained from the Jackson Laboratory Induced Mutant Resource. Mice carrying the conditional *Pten^flox* (B6.129S4-*Pten^tm1Hwu*/J) allele (*Lesche et al., 2002*) and the inducible *Kras^lsl-G12D* (B6.129-*Kras^tm4Tyj*/Nci) allele (*Jackson et al., 2001*) were obtained from the National Cancer Institute Mouse Models of Human Cancer Consortium Repository. Mice with the conditional *Ar^flox* (*B6N.129-Ar^tm1Verh*/Cnrm) allele (*De Gendt et al., 2004*) was obtained from the European Mouse Mutant Archive. Animals were maintained on a congenic C57BL/6N background. Genotyping was performed using the primers listed in *Supplementary file 1A*. Primer sequences used for genotyping of *Ar* alleles were previously described (*Yeh et al., 2003*).

## Mouse procedures

For lineage-marking and simultaneous deletion of AR in CARNs, *Nkx3.1^CreERT2/+*; *Ar^flox/Y*; *R26R-YFP/+* males were castrated at 8 weeks of age and allowed to regress for 4 weeks, followed by administration of tamoxifen (Sigma; 9 mg/40 g body weight in corn oil) by daily oral gavage for four consecutive days, and a chase period of 4 weeks. Administration of testosterone for prostate regeneration (Sigma; 25 mg/ml in 100% ethanol and diluted in PEG-400 to a final concentration of 7.5 mg/ml) was performed by subcutaneous implantation of mini-osmotic pumps (Alzet) that release testosterone solution at a rate of 1.875 µg/hr, which yields physiological levels of serum testosterone (*Banach-Petrosky et al., 2007*). For BrdU incorporation experiments, BrdU (Sigma; 100 mg/kg) was administered by intraperitoneal injection twice daily for 4 consecutive days, either from days 1 through 4 or from days 11 through 14 during androgen-mediated regeneration.

For cell of origin experiments, *Nkx3.1^CreERT2/+*; *Pten^flox/flox*; *Ar^flox/Y*; *R26R-YFP/+* and *Nkx3.1^CreERT2/+*; *Pten^flox/flox*; *Kras^LSL-G12D/+*; *Ar^flox/Y*; *R26R-YFP/+* mice as well as corresponding controls were castrated at 8 to 12 weeks of age. One month later, mice were administered tamoxifen, with a chase period of 3 months, followed by androgen-mediated regeneration for 1 month; mice were then euthanized for analysis. All animal experiments were performed according to protocols approved by the Institutional Animal Care and Use Committee at Columbia University Medical Center.

## Benign human prostate specimens

Radical prostatectomy samples were obtained from consented patients under the auspices of an Institutional Review Board approved protocol at Columbia University Medical Center. Tissue from benign prostate regions was dissected and transported to the laboratory in DMEM/F12 (Gibco) supplemented with 5% FBS. Benign pathology was first determined by H and E-staining of snap-frozen sections, and subsequently confirmed by immunostaining of paraffin sections for p63 and AMACR.

## Tissue acquisition, dissociation and isolation of prostate epithelial cells

Tissue dissociation and isolation were performed as previously described (*Chua et al., 2014*). In brief, mouse prostate tissue from all lobes was dissected in cold phosphate buffered saline (PBS) and minced with scissors. For human prostate specimens, tissue was cut into small pieces with scalpels, washed with PBS with 4 mg/ml Gentamicin (Gibco), and then minced with scissors. Both mouse and human prostate tissues were then incubated in DMEM/F12 (Gibco) supplemented with 5% FBS and 1:10 dilution of collagenase/hyaluronidase (STEMCELL Technologies) at 37°C for 3 hr. Dissociated tissues were spun at 350 g for 5 min, and resuspended in ice-cold 0.25% trypsin-EDTA (STEMCELL Technologies), followed by incubation at 4°C for 1 hr. Trypsinization was stopped by addition of Modified Hank's Balanced Salt Solution (HBSS) (STEMCELL Technologies) supplemented with 2% FBS. After centrifugation at 350 g, pelleted cells were resuspended with pre-warmed 5 mg/ml dispase (STEMCELL Technologies) supplemented with 1:10 dilution of 1 mg/ml DNase I (STEMCELL Technologies), triturated vigorously for 1 to 2 min, and diluted by addition of HBSS/2% FBS. Finally, the cell suspension was passed through a 40 µm cell strainer (Falcon).

## Flow cytometry

For flow sorting of mouse prostate epithelial cells, cell suspensions were stained on ice for 25 min with fluorescent-tagged EpCAM (BioLegend #118214) antibody. For isolation of human prostate epithelial cells, we used fluorescent-tagged EpCAM (BioLegend #324208, specific for human) and E-cadherin (eBioscience #46-3249-82) antibodies. The stained cells were spun, and cell pellets washed with HBSS/2% FBS, followed by resuspension in HBSS/2% FBS with 10 µM Y-27632 (ROCK inhibitor; STEMCELL Technologies) and a 1:1000 dilution of 0.5 mg/ml DAPI to exclude dead cells. Both side-scatter pulse width (SSC-W) vs. area (SSC-A) and forward side-scatter pulse area (FSC-A) vs. heights (FSC-H) were used to isolate single dissociated cells.

## Adherent culture for mouse and human prostate epithelial cells

To establish cell lines from lineage-marked CARNs as well as benign prostate epithelial cells, we performed adherent culture in our prostate organoid medium (*Chua et al., 2014*), consisting of hepatocyte medium supplemented with 10 ng/ml epidermal growth factor (EGF) (Corning), 10 µM Y-27632 (STEMCELL Technologies), 1x glutamax (Gibco), 5% Matrigel (Corning), 5% charcoal-stripped FBS (Gibco) heat-inactivated at 55°C for 1 hr, and supplemented with either 100 nM or 1 nM DHT (Sigma) for mouse and human cells, respectively. To derive APCA and ADCA lines, single YFP$^+$ cells from castrated and tamoxifen-treated $Nkx3.1^{CreERT2/+}$; $Ar^{flox/Y}$; $R26R$-$YFP/+$ mice were flow-sorted directly into 96-well Primaria plates (Corning), and were monitored daily to assess colony formation. Successful colonies were expanded and genotyped to assess the status of the $Ar^{flox}$ allele. For derivation of lines from benign human prostate epithelium, cells expressing either EpCAM and/or E-cadherin were plated into six-well Primaria plates at a density of 100,000 cells/well.

Passaging of adherent cultures was performed by removal of accumulated Matrigel on surface of the cells by gentle washing. The cells were washed with cold PBS, treated with 0.25% trypsin for 5 min at 37°C, and mechanically dissociated. Medium was changed every 4 days. Adherent cells were frozen in media consisting of 80% FBS, 10% complete medium, and 10% DMSO. Each APCA and ADCA line has been propagated continuously for at least eight passages.

## Cell culture assays

To assess cell viability, APCA and ADCA lines were plated in 96-well Primaria plates at a density of 1000 cells/well in the presence or absence of DHT. Cell viability was assayed at days 1, 2, 4 and 6 after plating using CellTiter-Glo 3D (Promega), with five technical replicates for each time point. In brief, CellTiter-Glo 3D reagent was thawed at 4°C and brought to room temperature prior to use. 100 µl of the reagent was added into each well containing 100 µl of medium. After shaking for 5–10 min, the mixture was then transferred to a 96-well CELLSTAR plate (Greiner), followed by incubation at room temperature for 10 min prior to measurement using a luminometer plate reader.

To assess colony formation, APCA and ADCA lines were plated in six-well Primaria plates at a density of 500 cells/well and grown for 9 days. three technical replicates were performed for each line in the presence or absence of DHT. At day 10 after plating, wells were washed with PBS and fixed with 100% methanol for 5 min. The wells were then washed with PBS for three times before staining with filtered 0.1% crystal violet solution. After drying the plates, colonies were counted, with a colony defined as a cell cluster containing at least 50 cells.

## Tissue recombination and renal grafting

For tissue recombination, 10 YFP$^+$ cells from castrated and tamoxifen-treated $Nkx3.1^{CreERT2/+}$; $Ar^{flox/Y}$; $R26R$-$YFP/+$ mice or control $Nkx3.1^{CreERT2/+}$; $R26R$-$YFP/+$ mice were combined with 250,000 dissociated rat urogenital mesenchyme cells from embryonic day 18.5 embryos, and resuspended in 15 µl of 9:1 collagen:setting buffer solution (10x Earle's Balanced Salt Solution (Life Technologies), 0.2 M NaHCO$_3$, and 50 mM NaOH). The recombinants were cultured overnight in DMEM with 10% FBS and 100 nM DHT, followed by grafting under the kidney capsules of male NOD.Cg-$Prkdc^{scid}$ $Il2rg^{tm1-Sug}$/JicTac (NOG) mice (Taconic). Renal grafts were harvested for analysis at 7–12 weeks after grafting. For the experiment involving APCA and ADCA lines, 100,000 cells were recombined with 250,000 rat urogenital mesenchyme cells, followed by grafting. At 6 weeks after grafting, some mice implanted with APCA cells were treated with tamoxifen to induce $Ar$ deletion.

Grafts were harvested for analysis after 12 weeks of growth and analyzed in paraffin sections for the presence of ducts expressing YFP. (Note that ducts can also be formed by YFP⁻cells that are derived from contaminating rat urogenital epithelium due to incomplete separation from the urogenital mesenchyme.) Graft efficiency was calculated on the basis of the presence of YFP⁺ ducts in the grafts using control CARNs and on the presence of YFP⁺AR⁻ ducts in the grafts using AR-deleted CARNs.

## Histology and immunostaining

For cryosections, tissues were fixed in 4% paraformaldehyde in PBS at 4°C overnight, placed in 30% sucrose in PBS overnight, and transferred to 1:1 30% sucrose in PBS and OCT (Tissue-Tek) solution for at least 4 hr prior to embedding in OCT. For paraffin sections, tissues were fixed in 10% formalin for 1 to 2 days, depending on size of tissue, prior to processing and embedding. Hematoxylin-eosin staining was performed using standard protocols. For immunostaining, sections underwent antigen-retrieval by heating in citrate acid-based or tris-based antigen unmasking solution (Vector Labs) for 45 min. Primary antibodies were applied to sections and incubated at 4°C overnight in a humidified chamber. Alexa Fluors (Life Technologies) were used as secondary antibodies. In some cases, tyramide amplification (Life Technologies) or ABC Elite (Vector Labs) kits together with HRP-conjugated or biotinylated secondary antibodies and NovaRed kit were used for signal detection. For immuno-fluorescent staining of cells, 5000 adherent cells/well were seeded on a eight-well Lab-Tek Chamber Slide (Nunc), grown for 4–8 days, and fixed with 4% paraformaldehyde for 10 min. After washing the slides with 3 changes of PBS, immunostaining was performed as above without antigen retrieval. Details of antibodies used are provided in *Supplementary file 1B*.

Histological grading of mouse prostate phenotypes was performed according to (*Park et al., 2002*). For lineage-tracing experiments, quantitation of marker staining was performed by manual counting of cells from confocal images taken with a 40x objective.

## RNA sequencing and bioinformatic analysis

For RNA preparation, APCA and ADCA cell lines at passage 5 or 6 were grown to approximately 70–80% confluency in Primaria 6-well plates in the presence of DHT, and lysed in Trizol. Total RNA extraction was performed using the 'No Spin' method of the MagMAX-96 for Microarrays kit (Ambion). Library preparation and RNA sequencing was performed by the Columbia Genome Center using their standard pipeline. In brief, mRNA was enriched by poly-A pull-down, and library preparation was performed using an Illumina TruSeq RNA prep kit. Libraries were pooled and sequenced using an Illumina HiSeq2500 instrument, yielding approximately 30 million single-ended 100 bp reads per sample. RTA (Illumina) was used for base calling and bcl2fastq2 (version 2.17) for conversion of BCL to fastq format, coupled with adaptor trimming. Reads were mapped to the mouse genome (UCSC/mm10) using STAR (2.5.2b) and FeatureCounts (v1.5.0-p3).

RNA-seq data raw counts were normalized and the variance was stabilized using DESeq2 package (Bioconductor) in R-studio 0.99.902, R v3.3.0 (The R Foundation for Statistical Computing, ISBN 3-900051-07-0). Differential gene expression signatures were defined as a list of genes ranked by their differential expression between any two phenotypes of interest (e.g. APCA versus ADCA lines; CRPC-NE versus CRPC, etc.), estimated using a two-sample two-tailed Welch t-test (for $n \geq 3$) or fold-change (for $n < 3$). For comparison of a mouse gene signature with a human gene signature, mouse genes were mapped to their corresponding human orthologs based on the homoloGene database (NCBI). Signatures were compared using Gene Set Enrichment Analysis (GSEA) (*Subramanian et al., 2005*), with the significance of enrichment estimated using 1000 gene permutations. Pathway enrichment analysis was performed using the C2 database, which includes pathways from REACTOME (*Fabregat et al., 2016*), KEGG (*Ogata et al., 1999*), and BioCarta (http://www.biocarta.com/genes/allpathways.asp). Expression data are deposited in the Gene Expression Omnibus database under GSE99233.

## Statistical analyses

Statistical analysis was performed using the Statistical Package for the Social Sciences (SPSS). Data distribution was assessed by the Kolmogorov-Smirnov test. Arcsine transformation was performed on data with non-normal distribution. Two-sample two-tail Welch t-test or Fisher's Exact Test was

performed for comparison between two independent groups as appropriate. No statistical methods were used to pre-determine sample size, and experiments were not randomized; investigators were not blinded to allocation during experiments and outcome assessment.

## Acknowledgements

We thank Robert Cardiff for advice on histology and Cory Abate-Shen, Laura Crowley, Maximilian Marhold, Maho Shibata, and Roxanne Toivanen for insightful discussions and comments on the manuscript. Our studies used the CCTI Flow Cytometry Core (supported in part by the Office of the Director, National Institutes of Health under awards S10RR027050), and the HICCC Molecular Pathology Shared Resource. This work was supported by post-doctoral fellowships from the DOD Prostate Cancer Research Program (CWC, BIL), by a Rutgers SHP Dean's Research Intramural grant (AM), and by grants from the Prostate Cancer Foundation (MMS) and the National Institutes of Health (MMS).

## Additional information

### Funding

| Funder | Grant reference number | Author |
| --- | --- | --- |
| National Institute of Diabetes and Digestive and Kidney Diseases | DK076602 | Michael M Shen |
| National Cancer Institute | CA1966692 | Michael M Shen |
| U.S. Department of Defense | Prostate Cancer Research Program PC101820 | Chee Wai Chua |
| U.S. Department of Defense | Prostate Cancer Research Program PC141064 | Bo I Li |
| Prostate Cancer Foundation | | Michael M Shen |
| Rutgers SHP Dean's Intramural Grant | | Antonina Mitrofanova |

The funders had no role in study design, data collection and interpretation, or the decision to submit the work for publication.

### Author contributions

Chee Wai Chua, Conceptualization, Data curation, Formal analysis, Funding acquisition, Validation, Investigation, Visualization, Methodology, Writing—original draft; Nusrat J Epsi, Eva Y Leung, Formal analysis, Investigation, Visualization, Methodology; Shouhong Xuan, Ming Lei, Formal analysis, Investigation, Methodology; Bo I Li, Data curation, Formal analysis, Funding acquisition, Investigation, Methodology; Sarah K Bergren, Formal analysis, Supervision, Investigation, Methodology; Hanina Hibshoosh, Resources, Formal analysis, Investigation, Methodology; Antonina Mitrofanova, Data curation, Software, Formal analysis, Supervision, Funding acquisition, Investigation, Visualization, Methodology, Writing—original draft, Project administration; Michael M Shen, Conceptualization, Resources, Formal analysis, Supervision, Funding acquisition, Investigation, Visualization, Writing—original draft, Project administration, Writing—review and editing

### Author ORCIDs

Chee Wai Chua https://orcid.org/0000-0002-9634-7265
Nusrat J Epsi http://orcid.org/0000-0001-5363-075X
Eva Y Leung http://orcid.org/0000-0003-1291-6625
Shouhong Xuan http://orcid.org/0000-0003-0571-7855
Bo I Li http://orcid.org/0000-0002-4104-4179
Antonina Mitrofanova http://orcid.org/0000-0003-0671-6512
Michael M Shen http://orcid.org/0000-0002-4042-1657

## Ethics

Human subjects: Radical prostatectomy samples were obtained from consented patients under the auspices of an Institutional Review Board approved protocol AAAC4997 at Columbia University Medical Center.

Animal experimentation: All animal experiments were performed under protocol AAAR9408, which was approved by the Institutional Animal Care and Use Committee at Columbia University Medical Center.

## Decision letter and Author response

Decision letter https://doi.org/10.7554/eLife.28768.030
Author response https://doi.org/10.7554/eLife.28768.031

# Additional files

## Supplementary files

• Supplementary file 1. Primers and antibodies used in this study.
DOI: https://doi.org/10.7554/eLife.28768.015

• Transparent reporting form
DOI: https://doi.org/10.7554/eLife.28768.016

## Major datasets

The following dataset was generated:

| Author(s) | Year | Dataset title | Dataset URL | Database, license, and accessibility information |
|---|---|---|---|---|
| Chua, CW, Li BI, Mitrofanova, A, Shen MM | 2017 | Differential requirements of androgen receptor in luminal progenitors during prostate regeneration and tumor initiation (APCA and ADCA lines RNASeq) | https://www.ncbi.nlm.nih.gov/geo/query/acc.cgi?acc=GSE99233 | GSE99233 |

The following previously published datasets were used:

| Author(s) | Year | Dataset title | Dataset URL | Database, license, and accessibility information |
|---|---|---|---|---|
| Best CJ, Gillespie JW, Yi Y, Chandramouli GV, Perlmutter MA, Gathright Y, Erickson HS, Georgevich L, Tangrea MA, Duray PH, González S, Velasco A, Linehan WM, Matusik RJ, Price DK, Figg WD, Emmert-Buck MR, Chuaqui RF | 2005 | Molecular alterations in primary prostate cancer after androgen ablation therapy | https://www.ncbi.nlm.nih.gov/geo/query/acc.cgi?acc=GSE2443 | Publicly available at the NCBI Gene Expression Omnibus (accession no: GSE2443) |
| Stanbrough M, Bubley GJ, Ross K, Golub TR, Rubin MA, Penning TM, Febbo PG, Balk SP | 2006 | Increased expression of genes converting adrenal androgens to testosterone in androgen-independent prostate cancer | https://www.ncbi.nlm.nih.gov/geo/query/acc.cgi?acc=GSE32269 | Publicly available at the NCBI Gene Expression Omnibus (accession no: GSE32269) |

| Beltran H, Prandi D, Mosquera JM, Benelli M, Puca L, Cyrta J, Marotz C, Giannopoulou E, Chakravarthi BV, Varambally S, Tomlins SA, Nanus DM, Tagawa ST, Van Allen EM, Elemento O, Sboner A, Garraway LA9, Rubin MA, Demichelis F | 2016 | Divergent clonal evolution of castration-resistant neuroendocrine prostate cancer | https://www.ncbi.nlm.nih.gov/projects/gap/cgi-bin/study.cgi?study_id=phs000909.v1.p1 | Publicly available at dbGaP (accession no. dbGaP phs000909.v1.p1) |
| Zhang D, Park D, Zhong Y, Lu Y, Rycaj K, Gong S, Chen X, Liu X, Chao HP, Whitney P, Calhoun-Davis T, Takata Y, Shen J, Iyer VR, Tang DG | 2016 | Stem cell and neurogenic gene-expression profiles link prostate basal cells to aggressive prostate cancer | https://www.ncbi.nlm.nih.gov/geo/query/acc.cgi?acc=GSE67070 | Publicly available at the NCBI Gene Expression Omnibus (accession no: GSE67070) |
| Xie Q, Liu Y, Cai T, Horton C, Stefanson J, Wang ZA | 2017 | Dissecting cell-type-specific roles of androgen receptor in prostate homeostasis and regeneration through lineage tracing | https://www.ncbi.nlm.nih.gov/geo/query/acc.cgi?acc=GSE76724 | Publicly available at the NCBI Gene Expression Omnibus (accession no: GSE76724) |

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
