## [Decision Letter]

Thank you for submitting your article "Differential requirements of androgen receptor in luminal progenitors during prostate regeneration and tumor initiation" for consideration by *eLife*. Your article has been reviewed by three peer reviewers, and the evaluation has been overseen by Sean Morrison as the Senior and Reviewing Editor. The reviewers have opted to remain anonymous.

The reviewers have discussed the reviews with one another and the Reviewing Editor has drafted this decision to help you prepare a revised submission.

Androgen receptor is a major regulator essential for prostate development and prostate cancer progression, but its specific roles in prostate stem/progenitor cells are not fully clarified. By using a previous established CARNs mouse model, the current study investigated AR functions in luminal prostate progenitor cells. Although deletion of AR did not affect the progenitor properties of CARNs, loss of AR suppressed tumor formation from PTEN-deleted CARNs. Paradoxically, however, loss of AR promoted tumor aggressiveness in a PTEN-/-Kras+/+ model. Overall, the studies are very well designed and the data are solid. Together, the findings imply that AR may modulate luminal progenitor cells properties including their ability to serve as a cell of origin of prostate cancer.

1) One major limitation in the study of CARN's is that they can only be defined in the castrated mouse. One cannot label the luminal cell that will survive castration and become the CARN without castration. Thus, all labeling as well as activation of tumorigenic events is done in the castrated state. This is caveat should be explicitly noted in the paper.

2) Comparing this work with other recent work, one important question is whether and how CARN's are different from bulk luminal cells. AR deletion in bulk luminal cells, whether using Pb-Cre4 or Nkx3-1-CreERT2 in the non-castrated state does not decrease the ability to regress and regenerate or decrease Pten-loss mediated tumorigenesis (Xie, 2017). However, AR deletion in CARN's seems to be a detriment to regeneration and to Pten-loss mediated tumorigenesis. As noted above though, is the difference due to a different luminal cell (CARN vs. bulk) or Pten deletion in the castrated vs. intact state? This question could be answered using a luminal CreERT2 (CK8 or CK18) in castrated or intact mice during tamoxifen injection to delete Pten. If the authors have these mice it would be very helpful to address this. If not, they should discuss this issue.

3) What is the relative efficiency between AR+ and AR- CARN's in establishing 2D cultures? The authors used AR-floxed mice and compared clones that per chance were AR-deleted vs. intact.

4) The data demonstrating that AR depletion in PTEN/KRAS results in aggressive NE tumors is not convincing. The data describing "the important difference between the NPK-CARN and NPKA-CARN tumors" is weak and based on "NPKA-CARN tumors contained a low percentage of synaptophysin-positive neuroendocrine cells among total epithelial cells (0.7%, n=3), which was not observed in the NPK-CARN controls (0%, n=3) (Figure 6). Since these neuroendocrine cells express YFP (Figure 6), we conclude that transformed AR-negative CARNs can give rise to neuroendocrine cells".

The authors need to do more than this to show focal NE lesions. At least 3 NE markers should be used and the incidence presented more frankly. Being as low as 0.7%, these cells were difficult to detect…did all 3 mice have focal NE lesions? The error bars in Figure 6 are large and suggest that maybe not all mice showed detectable lesions. This is important to know. If this is to be heralded as a new model, a more thorough presentation of the pathology is required/essential. There is no data in the paper that show these tumors were more aggressive and the authors should tone down the claim "that AR down-regulation primes CARNs to serve as a cell of origin for aggressive CRPC with focal neuroendocrine differentiation". The differentiation of NE cells in prostate tumors is a very hot topic and very confusing right now. It would not be helpful to add to the confusion.

5) The Introduction focuses entirely on murine prostate stem and progenitor cells with no discussion of the similarities and, importantly, the differences relative to human prostate stem and progenitors. This is problematic in that the authors switch over to the relevance of the issues raised in the present experiments to human prostate cancer, CRPC, and lineage switching to the neuroendocrine phenotype. While unipotent progenitors have been shown in the murine models for basal and luminal progenitors, lineage tracing in human prostate identifies a common precursor for basal, luminal and neuroendocrine cells (e.g. Blackwood, 2011, Gaisa, 2011). In contrast, there is some, but rather limited evidence for CARNs in human prostate. As such, the authors need to expand on these aspects in the present paper and discuss the possibility that the results in the paper may not be directly translated to the human system.

---

## [Author Response]

1) One major limitation in the study of CARN's is that they can only be defined in the castrated mouse. One cannot label the luminal cell that will survive castration and become the CARN without castration. Thus, all labeling as well as activation of tumorigenic events is done in the castrated state. This is caveat should be explicitly noted in the paper.

As noted by the reviewer(s), CARNs are defined in the regressed prostate epithelium, and thus all of the in vivoexperiments involve manipulations performed following castration. We agree that this point should be clearly stated, and have added this caveat to the fifth paragraph of the Discussion.

2) Comparing this work with other recent work, one important question is whether and how CARN's are different from bulk luminal cells. AR deletion in bulk luminal cells, whether using Pb-Cre4 or Nkx3-1-CreERT2 in the non-castrated state does not decrease the ability to regress and regenerate or decrease Pten-loss mediated tumorigenesis (Xie, 2017). However, AR deletion in CARN's seems to be a detriment to regeneration and to Pten-loss mediated tumorigenesis. As noted above though, is the difference due to a different luminal cell (CARN vs. bulk) or Pten deletion in the castrated vs. intact state? This question could be answered using a luminal CreERT2 (CK8 or CK18) in castrated or intact mice during tamoxifen injection to delete Pten. If the authors have these mice it would be very helpful to address this. If not, they should discuss this issue.

As discussed in the third and fourth paragraphs of the Discussion, there are several lines of evidence that CARNs differ from bulk luminal cells in the regressed prostate epithelium. However, as noted by the reviewer(s), the effect of AR deletion in CARNs on tumor formation following *Pten* inactivation could in principle be due to differences between CARNs and bulk luminal cells and/or differences due to *Pten* loss in the regressed versus hormonally-intact prostate epithelium. The suggested experiment using an inducible luminal cytokeratin Cre driver for *Pten* deletion in castrated versus intact mice might help distinguish these possibilities, but we believe that this experiment lies beyond the scope of the present study. However, we have revised the Discussion to explicitly discuss this caveat for our interpretation in the Discussion.

3) What is the relative efficiency between AR+ and AR- CARN's in establishing 2D cultures? The authors used AR-floxed mice and compared clones that per chance were AR-deleted vs. intact.

In response to this comment, we have added new experimental data that directly compares the efficiency of AR^+^ and AR^–^ CARNs in establishing 2-dimensional cell lines. In brief, we sorted 60 single YFP^+^ cells from castrated and tamoxifen-treated *Nkx3.1^CreERT2/+^; Ar^flox/Y^; R26RYFP/+* experimental mice into wells of a 96-well plate. We found that 6 cells gave rise to adherent lines, with 4 of these being AR^+^ lines that had failed to undergo Cre-mediated recombination, and 2 lines being AR^–^ lines. After correcting for the observed 87.1% efficiency of recombination of the AR-floxed allele in vivo(Results, eighth paragraph), the relative efficiency for the AR^–^ CARNs compared to AR^+^ CARNs is 7.4%. The decreased plating efficiency of AR^–^ CARNs is also consistent with the tissue grafting data in Figure 3. This experiment is now described in the Results section.

4) The data demonstrating that AR depletion in PTEN/KRAS results in aggressive NE tumors is not convincing. The data describing "the important difference between the NPK-CARN and NPKA-CARN tumors" is weak and based on "NPKA-CARN tumors contained a low percentage of synaptophysin-positive neuroendocrine cells among total epithelial cells (0.7%, n=3), which was not observed in the NPK-CARN controls (0%, n=3) (Figure 6). Since these neuroendocrine cells express YFP (Figure 6), we conclude that transformed AR-negative CARNs can give rise to neuroendocrine cells".The authors need to do more than this to show focal NE lesions. At least 3 NE markers should be used and the incidence presented more frankly. Being as low as 0.7%, these cells were difficult to detect…did all 3 mice have focal NE lesions? The error bars in Figure 6 are large and suggest that maybe not all mice showed detectable lesions. This is important to know. If this is to be heralded as a new model, a more thorough presentation of the pathology is required/essential. There is no data in the paper that show these tumors were more aggressive and the authors should tone down the claim "that AR down-regulation primes CARNs to serve as a cell of origin for aggressive CRPC with focal neuroendocrine differentiation". The differentiation of NE cells in prostate tumors is a very hot topic and very confusing right now. It would not be helpful to add to the confusion.

We agree with the reviewers that our description of the focal neuroendocrine differentiation in the *NPKA-CARN* tumors was insufficiently detailed. As suggested by the reviewers, we have provided additional quantitation of the synaptophysin-positive cells in the *NPKA-CARN* tumors. These data are now presented for each individual tumor in Figure 6—source data 1, which shows that all three tumors contained synaptophysin-positive cells, but to varying degrees, resulting in the relatively large error bar in Figure 6. We have also added panels in Figure 6 showing expression of additional neuroendocrine markers, corresponding to Foxa2 and Chromogranin A, which were observed in all three tumors, supporting the occurrence of focal neuroendocrine differentiation in *NPKA-CARN* tumors. Furthermore, we agree that the text in the Discussion was previously overstated, and have now edited the text to limit our claims to the occurrence of neuroendocrine transdifferentiation in *NPKA-CARN* tumors.

5) The Introduction focuses entirely on murine prostate stem and progenitor cells with no discussion of the similarities and, importantly, the differences relative to human prostate stem and progenitors. This is problematic in that the authors switch over to the relevance of the issues raised in the present experiments to human prostate cancer, CRPC, and lineage switching to the neuroendocrine phenotype. While unipotent progenitors have been shown in the murine models for basal and luminal progenitors, lineage tracing in human prostate identifies a common precursor for basal, luminal and neuroendocrine cells (e.g. Blackwood, 2011, Gaisa, 2011). In contrast, there is some, but rather limited evidence for CARNs in human prostate. As such, the authors need to expand on these aspects in the present paper and discuss the possibility that the results in the paper may not be directly translated to the human system.

We agree that discussion of epithelial progenitors and the potential existence of CARNs in the human prostate is relevant for the interpretation of our findings, and therefore have added a paragraph on this topic in the Discussion. While functional lineage-tracing approaches as performed in genetically-engineered mice are not feasible in human prostate in vivo, lineage-reconstruction studies by analysis of patterns of mitochondrial DNA mutations have yielded insights into potential lineage relationships. As mentioned by the reviewer(s), previous studies have indicated the existence of multipotent epithelial progenitors (Gaisa et al., 2011; Blackwood et al., 2011). Furthermore, recent analyses have provided evidence for multipotent basal progenitors localized to the most proximal region of the prostate as well as more distally located unipotent luminal progenitors (Moad et al.,). Notably, there is also evidence for bipotential luminal progenitors in ex vivostudies of human prostate organoids (Karthaus et al., 2014). Our revised manuscript also cites evidence for the existence of CARN-like cells in studies of human prostate tissue slice culture (Zhao et al.,2010) and prostate tumors (Germann et al.,2012). Thus, we believe that current data favor a general similarity of epithelial lineage relationships in the two species, suggesting that findings deduced from analyses of the mouse prostate may be translatable to the human prostate. These points are now discussed in the fifth paragraph of the Discussion.